# Definition of functionally and structurally distinct repressive states in the nuclear receptor PPARγ

Zahra Heidari[1,2,5], Ian M. Chrisman[2,3,5], Michelle D. Nemetchek [2,3], Scott J. Novick[4], Anne-Laure Blayo[4], Trey Patton[1], Desiree E. Mendes[1], Philippe Diaz[1], Theodore M. Kamenecka[4], Patrick R. Griffin [4] & Travis S. Hughes [1,2,3]*

The repressive states of nuclear receptors (i.e., apo or bound to antagonists or inverse agonists) are poorly defined, despite the fact that nuclear receptors are a major drug target. Most ligand bound structures of nuclear receptors, including peroxisome proliferator-activated receptor γ (PPARγ), are similar to the apo structure. Here we use NMR, accelerated molecular dynamics and hydrogen-deuterium exchange mass spectrometry to define the PPARγ structural ensemble. We find that the helix 3 charge clamp positioning varies widely in apo and is stabilized by efficacious ligand binding. We also reveal a previously undescribed mechanism for inverse agonism involving an omega loop to helix switch which induces disruption of a tripartite salt-bridge network. We demonstrate that ligand binding can induce multiple structurally distinct repressive states. One state recruits peptides from two different corepressors, while another recruits just one, providing structural evidence of ligand bias in a nuclear receptor.

[1] Department of Biomedical and Pharmaceutical Sciences, University of Montana, 32 Campus Drive, Missoula, MT 59812, USA. [2] Center for Biomolecular Structure and Dynamics, University of Montana, 32 Campus Drive, Missoula, MT 59812, USA. [3] Biochemistry Graduate Program, University of Montana, 32 Campus Drive, Missoula, MT 59812, USA. [4] Department of Molecular Medicine, The Scripps Research Institute, 120 Scripps Way, Jupiter, FL 33458, USA. [5] These authors contributed equally: Zahra Heidari, Ian M. Chrisman. *email: travis.hughes@umontana.edu

The nuclear receptor superfamily of transcription factors modulate transcription of myriad genes implicated in development, immunity, metabolism, and reproduction[1], and are a major drug target, accounting for ~16% of all approved drugs[2]. While these drugs provide unique benefits for many conditions, their utility is often limited by undesired effects. For example, drugs that target the nuclear receptors glucocorticoid receptor or PPARγ cause bone loss in addition to valuable anti-inflammatory or anti-diabetes effects[3–5]. Some experimental drugs, often termed selective nuclear receptor modulators, produce less undesired effects, but similar desired effects as prescribed drugs[6]. The underlying mechanism of lessened undesired effects may involve drugs inducing a distinct receptor structural state from current drugs, an idea known as ligand bias[7]. Confirming drug specific structure and connection of that structure to function is essential for future biased nuclear receptor drug development.

Most nuclear receptors share similar mechanisms of modulation by drugs[8]. Nuclear receptors share a common structural architecture, including modular ligand-binding domains (LBD) and DNA-binding domains that are connected by a flexible hinge region (Fig. 1a). About half of the 48 human nuclear receptors are known to heterodimerize with RXR, including PPARγ; however, PPARγ has been found as both monomers and heterodimers in cells leaving open the possibility that it and other RXR partners can signal as monomers[9]. Drugs bind deep in a ligand binding pocket within the LBD and allosterically change the receptor surface. Changes to receptor surfaces that interface with coregulator proteins are especially important because bound coregulators affect gene expression via chromatin modification, bridging to transcriptional machinery, or other mechanisms[6,10,11]. Coregulators are often classified as either coactivators or corepressors which increase or decrease gene transcription respectively. Short (~20 residue) helical regions in coregulators, termed LxxLL and CoRNR boxes, mediate binding to nuclear receptors[12,13]. We refer herein to the receptor conformation in coactivator LxxLL box peptide bound structures as the active state.

Models of drug induced changes to the coregulator binding surface are based on the over 800 nuclear receptor structures[14], simulations[15], nuclear magnetic resonance (NMR)[16,17], fluorescence anisotropy[18–20], and hydrogen deuterium exchange mass spectrometry (HDX-MS)[21,22]. These reports support the idea that the position and/or dynamics of the c-terminal helix in most nuclear receptors (helix 12) is an important determinant of activity. For example, helix 12 is found in distinct conformations in apo/antagonist versus agonist bound receptors for some structures of RXRα, ERα, and PPARα, supporting a structure–function model with a few distinct states[14,23]. However, the lack of such drug-induced changes for other receptors and solution state data has led to the dynamic stabilization model[8], which posits that activation involves reduction of helix 12 movement and/or an increase in helical integrity. Consistent with this model, protein NMR shows that agonist binding diminishes intermediate exchange (i.e., μs–ms dynamics) throughout the LBD[16,17]. On the other hand, absence of helix 12 structural changes in some crystal structures may misrepresent the physiologic structural ensemble because the dominant structure in solution may not be readily crystallizable. For example, the ~200 PPARγ crystal structures lack differences in helix 12 conformation despite our recent fluorine NMR data, which shows ligand dependent distinct conformations[24].

Repressive nuclear receptor states, which favor corepressor and/or disfavor coactivator binding, and the mechanisms by which ligands induce these states are even less defined than for active states. The basic mechanism of ligand induced discrimination between coactivator and corepressor is known. The corepressor CoRNR box[25] helix is longer than the coactivator LxxLL box[12] helix. The longer CoRNR helix requires some helix 12 displacement from the canonical active state and/or loss of helix 12 structure for optimal binding. Both helix 12 displacement and/or loss of helical integrity has been observed in structures of PPARα[26], PR[27], ERRγ[28], RORγ[29], RARα[19], and GR[30] bound to CoRNR boxes. Insight regarding repressive structural states using non-crystallographic methods is limited[21,31]. No PPARγ-corepressor structure has been published; however, we recently used fluorine NMR to show that PPARγ helix 12 is found in two distinct repressive structural states (apo and inverse agonist bound). These two states are different from the agonist bound helix 12 state[24], which is the only currently well-defined state.

The data presented here (simulations, mutagenesis, protein and $^{19}$F NMR, and HDX-MS) reveals atomic resolution details for several repressive states of PPARγ. These data support the idea that efficacious ligands induce both large scale structural changes, as in the mouse-trap model[32], and stabilization of dynamics, including increases in helix 12 integrity, as envisioned in a dynamic stabilization model[8]. We show that two inverse agonists induce distinct structural ensembles from apo PPARγ and from each other via distinct mechanisms (we use the term induce in this work in the general sense; not to imply the absence of conformational selection[33]). We first detail the apo ensemble of structures in regions of PPARγ that comprise the coregulator binding surface (helix 3 and 12). We then demonstrate that different inverse agonists induce distinct changes to this apo ensemble including changes to the helix 3 charge clamp. Next, we reveal an undescribed molecular mechanism of inverse agonism involving disruption of a tripartite salt bridge. These changes are induced by one inverse agonist, but not another, via elongation of helix 3. Finally, we demonstrate that these distinct structural ensembles yield distinct functional effects through precise measurement of PPARγ affinity for CoRNR peptides from nuclear receptor corepressor 1 (NCoR) and 2 (SMRT). These data demonstrate that drugs can induce structurally and functionally distinct repressive states in a nuclear receptor. This is a step beyond the current rheostat model of nuclear receptor structure–function toward a conceptual framework that explains in vitro and in vivo evidence of ligand bias[7] in nuclear receptors[6].

## Results

**Labeling PPARγ for fluorine NMR.** To improve the nuclear receptor structure–function model we gathered information regarding the number and population of structures that compose the PPARγ LBD ensemble, the exchange rate between those structures, and the effect of drugs and coregulators on this ensemble using fluorine NMR. Fast exchange of the fluorine probe between chemical shift environments on the ps–ns time-scale indicates local movement (e.g., side chain rotamer jumps) and produces one narrow peak at the average of the visited chemical shifts[34,35]. As exchange between conformations slows, NMR signals broaden and eventually wide individual peaks begin to emerge (i.e., intermediate exchange). Residues in intermediate exchange (μs–ms lifetimes) can be detected using fluorine NMR, but are often undetectable/unresolvable in protein NMR. As lifetimes increase toward seconds, peaks narrow and move toward their individual chemical shifts (i.e., slow exchange, Fig. 1c). The slower the exchange rate, the more likely the coordinated movement involves many atoms, possibly the entire protein[34]. The relative area of peaks within a NMR spectrum can correspond to the relative populations of structural states that compose the overall ensemble.

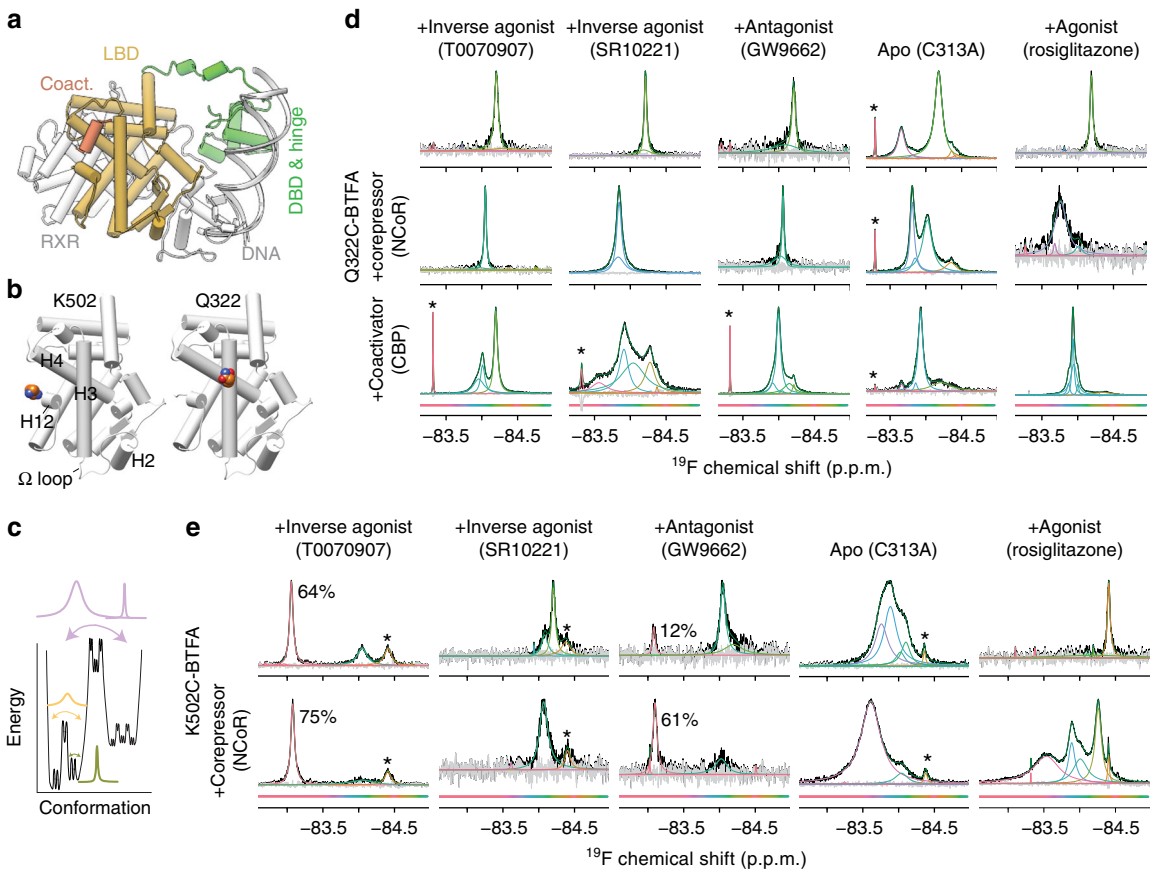

**Fig. 1 NMR indicates that the Helix 3 and helix 12 structural ensembles are diverse and ligand dependent. a** Overall architecture of a nuclear receptor signaling complex (PDB code 3DZY). The DNA-binding domain (DBD), hinge region, and ligand-binding domain (LBD) of PPARγ are shown bound to a coactivator peptide (coact.) and DNA. The heterodimerization partner of PPARγ is also shown (RXR). **b** Location of residues that were changed to cysteine and labeled with BTFA (colored spheres) on the PPARγ LBD for the two variants used in panels (**d**) and (**e**). Helix 2, 3, 4, and 12 and the omega loop (Ω loop) are indicated. **c** Illustration of the relationship between energy of a structural conformation, barrier height between conformations, and ¹⁹F NMR spectra. Color denotes the exchange rate between conformations in the energy wells. Purple denotes slow exchange (e.g., ms–s lifetime), tan denotes intermediate exchange (e.g., μs–ms lifetime), and olive denotes fast exchange (ps–ns lifetime). **d**, **e** ¹⁹F NMR spectra of PPARγ labeled on helix 3 (panel **d**) and helix 12 (panel **e**) with BTFA and bound to the indicated ligands. Individual deconvoluted peaks are colored according to chemical shift as indicated by the color bars below the spectra. The lower rows of spectra include a 2× molar ratio of CoRNR box peptides derived from the corepressor NCoR or the coactivator CREB-binding protein (CBP). The percentages in panel e refer to the portion of the spectral area found in the left peak. Asterisk denotes unfolded PPARγ, free BTFA, or BTFA labeled contaminating protein[24].

To aid interpretation of our spectra, we extract the probable number, location, relative population, and width of overlapping peaks using a deconvolution program which produces the most parsimonious fit that accurately describes the spectrum using a Bayesian information criteria score[36].

To enable ¹⁹F NMR of the PPARγ LBD we attached a trifluoromethyl NMR probe (BTFA[37]; Supplementary Fig. 14b) to an introduced cysteine on either helix 12 (PPARγ$^{K502C}$-BTFA), helix 3 (PPARγ$^{Q322C}$-BTFA), or the omega loop (PPARγ$^{Q299C}$-BTFA). We use PPARγ2 isoform residue numbering throughout this work. Each PPARγ construct has the probe in only one location (Fig. 1b). We chose these locations as helix 12 and helix 3 form most of the surface that contacts coregulators and the omega loop conformation may affect receptor activity[23,38]. We label in the presence of bound ligand, or for apo we use a C313A mutant, to avoid labeling of the only native cysteine, which points into the ligand binding pocket. We had found previously that helix 12 labeling did not significantly change MED1, NCoR, and SMRT affinity except for apo and agonist (GW1929) bound PPARγ$^{K502C}$-BTFA which showed 4.5- and 1.4-fold increased affinity, respectively, for SMRT[24]. We tested the effect of the other two labels used here and found that omega loop labeling had little

effect (Supplementary Table 1), while helix 3 (Q322C) labeling increased affinity for MED1 1.7-fold and did not change affinity for NCoR (Supplementary Table 2). Apo PPARγ$^{C313A,Q322C}$-BTFA has a 1.7-fold decreased affinity for MED1 and 1.4-fold increased affinity for NCoR (Supplementary Table 2).

**Ligands induce distinct helix 12 structural states.** To connect structure to function in this work, we primarily use two inverse agonists (T0070907 and SR10221[21]), one antagonist (GW9662[39]), and an agonist (rosiglitazone). T0070907 and GW9662 covalently attach to the native cysteine and differ by one atom (Supplementary Fig. 14b). This difference converts T0070907 into an efficacious inverse agonist[31]. The antagonist and one inverse agonist (T0070907) increase affinity for CoRNR peptides from two corepressors (SMRT and NCoR), while the other inverse agonist (SR10221) increases affinity for SMRT only (Supplementary Tables 3 and 4). Only T0070907 and SR10221 reduce basal expression in a cell-based reporter assay[31], and thus we classify these as inverse agonists. ¹⁹F NMR of PPARγ$^{K502C}$-BTFA LBD confirmed our previous observations[24,31] that the inverse agonists and the antagonist induce distinct helix 12 structural ensembles from each other and from apo (Fig. 1e).

The GW9662 (antagonist) and T0070907 (inverse agonist) NMR spectra are similar, except that T0070907 has a larger left shifted peak and addition of corepressor peptides increase the population of this peak (Fig. 1e and Supplementary Fig. 10). This left-shifted peak is relatively narrow (43 Hz) indicating that it arises from very similar structures in relatively fast exchange. These data indicate that this left-shifted peak represents a helix 12 structural state with high affinity for corepressors.

**Ligands consolidate the complex apo helix 3 structural state.** The helix 3 charge clamp is a conserved lysine or arginine (K329 in PPARγ) that bonds to coactivators[40] and corepressors[26] and is important to corepressor affinity (Supplementary Figs. 11 and 12). Previously published HDX-MS data indicates that agonists increase helical integrity near this charge clamp[22]. We hypothesized that the structural state in this region of helix 3, like helix 12, is also ligand dependent and plays a role in coregulator affinity. Our previously published $^{19}$F NMR of PPARγ$^{C313A,Q322C}$-BTFA LBD demonstrated that the helix 3 structural ensemble is affected by ligands; however, the C313A mutation precludes T0070907 and GW9662 binding and affects non-covalent ligand binding and NCoR affinity[24]. We therefore performed additional experiments here using a PPARγ$^{Q322C}$-BTFA LBD construct, which has smaller effects on coregulator affinity and can covalently bind GW9662 and T0070907 (Fig. 1d and Supplementary Table 2).

Chemical exchange saturation transfer (CEST)[41] indicates the two broad apo peaks exchange less than 0.5 s$^{-1}$ for PPARγ$^{Q322C}$-BTFA LBD (Supplementary Fig. 3f, g), similar to what we previously observed for PPARγ$^{C313A,Q322C}$-BTFA LBD[24]. Apo peak broadening is not due to exchange between the left and right states as the ratio of chemical shift separation between the states (350 Hz) to the rate of exchange between states (<0.5 s$^{-1}$) is very small (<0.001)[36]. Broadening is also not due to exchange between monomer and homodimer forms of apo PPARγ. Small angle X-ray scattering and dynamic light scattering indicates PPARγ LBD is monomeric until at least 200 μM[17,42]. We performed fluorescence anisotropy of labeled PPARγ and did not detect homodimerization, furthermore helix 12 probe spectra of apo at 50, 150, and 300 μM appears identical (Supplementary Fig. 13). The two helix 3 apo structural states do not have grossly different affinity for NCoR or CBP as both peaks are in clear slow exchange with peptide bound peaks (Supplementary Fig. 3e). We routinely delipidate PPARγ, which binds lipids during purification from *Escherichia coli*[43]. The observed exchange between the two peaks could reflect binding and unbinding of lipid. Delipidation increases the population in the left apo peak, indicating that at a minimum, the left peak is lipid free (Supplementary Fig. 3b). The left apo peak is wider (~70 Hz) than the narrowest PPARγ-BTFA peaks we observe (~30–40 Hz) indicating that the left peak arises from two or more distinct structures exchanging on the μs to ms timescale. Overall, these data indicate that the helix 3 charge clamp region exchanges between at least two distinct structural states in apo PPARγ on the μs–s timescale.

All ligands collapse these broad apo peaks into primarily a single narrow peak (Fig. 1d), indicating that they all stabilize a mobile apo helix 3 charge clamp region. However, the antagonist and one inverse agonist (SR10221) show some evidence of persistent multiple states (Fig. 1d). Consistent with this NMR data, HDX-MS shows faster exchange in a peptide from a helix 3 region near the charge clamp (residues 320–326) for the SR10221 and apo complexes compared to the T0070907 containing complex (Supplementary Fig. 8). These data indicate that one inverse agonist (T0070907) and agonist (rosiglitazone) fully stabilize the mobile apo charge clamp while the other

inverse agonist and the antagonist do not. In addition these data suggest that the charge clamp position and dynamics are ligand dependent and functionally important to corepressor and coactivator binding.

We previously observed that addition of matching coregulators and drugs to PPARγ (i.e., addition of coactivators to agonist complexes), leads to minor spectral changes for the helix 12 fluorine probe. In contrast, addition of opposing coregulators (i.e., addition of a corepressor to agonist complexes) leads to peak broadening and peak splitting. A third outcome was noted for addition of coactivators or corepressors alone to apo PPARγ, where both had similar effects[24]. These data indicated that efficacious agonists and inverse agonists strongly induce helix 12 structural ensembles ideal for either coactivator or corepressor binding while the structurally diverse apo ensemble does not strongly favor either.

To determine if ligands also induce specific helix 3 structures that favor coactivator and/or corepressor binding we added coregulators to PPARγ$^{Q322C}$-BTFA LBD alone and complexed with various ligands. Heterodimerization with RXRα reduces the affinity of PPARγ bound to T0070907 for a corepressor (NCoR) and increases it for a coactivator (MED1)[24] and is expected to skew the conformational ensemble in a similar manner to coactivator binding. Binding of coactivators, corepressors, or RXRα all produce similar changes to the apo and partial agonist (nTZDpa) bound spectra (Supplementary Figs. 3 and 7), indicating that the helix 3 charge clamp area has no clear preference for binding coactivators or corepressors for apo or a non-efficacious (partial) agonist complex. In contrast these data indicate that efficacious agonists and inverse agonists induce helix 3 structural ensembles with preference for either corepressor (T0070907 and SR10221) or coactivator (rosiglitazone) binding (Fig. 1d and Supplementary Fig. 7).

While efficacious ligands induce one primary helix 3 conformation, minor helix 3 conformations are also detected in these spectra. We tested exchange of minor with major conformations using CEST in the antagonist GW9662 as it had relatively large minor populations. CEST demonstrates slow exchange (95% CI: 1.7–2.2 s$^{-1}$) between downfield (left shifted) minor states with the major state for PPARγ$^{Q322C}$-BTFA LBD bound to GW9662 with or without RXRα (Supplementary Fig. 7c–g). The detected exchange is not between bound and unbound states because at NMR concentrations (150/300 μM PPAR/RXR) the heterodimer, with a $K_d$ of less than 6 nM, is saturated (Supplementary Fig. 13b). Thus, while efficacious ligands shift the helix 3 charge clamp ensemble toward one primary low energy structure, other higher energy structures are still lowly populated.

**Structural definition of the apo structural state.** To build an atomic resolution model of the structural ensemble of a nuclear receptor LBD we ran extensive accelerated molecular dynamics (aMD) simulations of apo and agonist (rosiglitazone) bound PPARγ LBD (Supplementary Table 5) starting from similar active helix 12 conformations (Supplementary Fig. 14). Similar to results from adaptive biasing-force simulations[15], the agonist simulations produce one relatively narrow energy well, while the apo PPARγ energy landscape has a wider range of conformations that would be significantly populated at physiologic temperature (Fig. 2a, b). Unlike adaptive biasing-force simulations, aMD does not use reaction coordinates but instead smooths the potential energy landscape, enhancing sampling efficiency without restricting the conformational space explored[44]. These data indicate that our aMD simulations sample enough of the helix 12 structural ensemble to differentiate ligand bound from apo PPARγ.

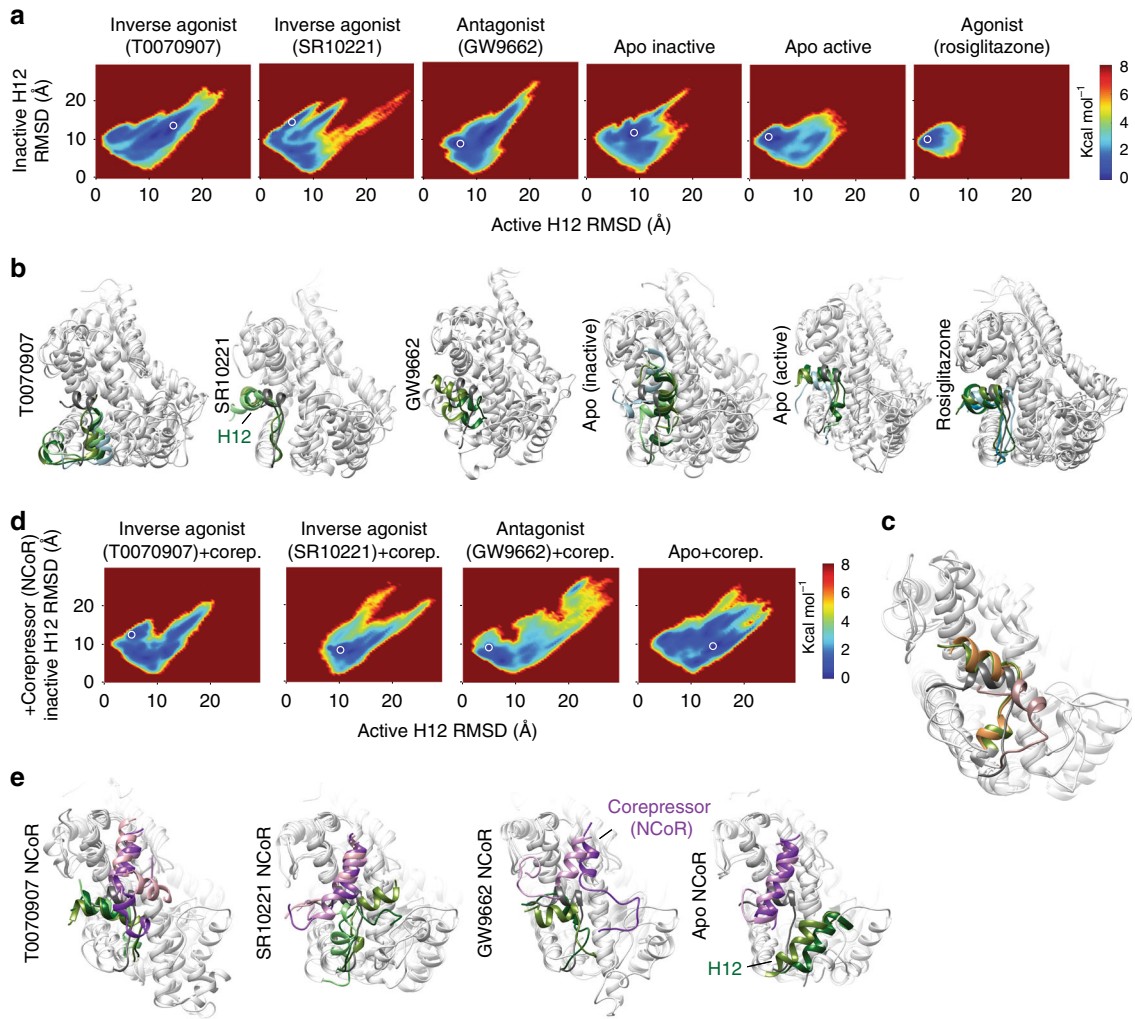

**Fig. 2 Extensive aMD simulations indicate a diverse PPARγ structural ensemble.** Multiple independent aMD simulations were run for the indicated PPARγ LBD complexes (see Supplementary Table 5). **a**, **d** The RMSD of helix 12 was calculated compared to active (chain A) and inactive (chain B) structures (PDB code: 1PRG). The energy of each trajectory snapshot was calculated and overlaid on the 2D RMSD to produce the displayed potential energy landscapes. **b**, **e** The aMD trajectory structures within a 0.2 × 0.2 Å RMSD square centered on the lowest energy wells (white circles in panels (**a**) and (**d**)) were clustered using k-means clustering into 5 clusters using CPPTRAJ[81]. Arbitrary (i.e., CPPTRAJ chosen) representative structures are shown for the most prevalent clusters. The relative prevalence of the structure is indicated by the color of helix 12 ranging from dark green for the most prevalent to olive drab, light green, light blue, and then deep sky blue for the least prevalent. For reference, an active structure (rosiglitazone bound PPARγ; 2PRG chain A) is shown with helix 12 colored dim gray. **c** Representative structures from the third and second lowest energy wells in the inactive and active apo simulations, respectively, are shown in orange and dark green. Active and inactive crystal structures are shown in dim gray and light pink, respectively.

We also started apo aMD simulations with helix 12 in an alternate crystal contact induced conformation[45], which we refer to as the inactive structure. This conformation is very different from the canonical active conformation; the helical axis of this inactive helix 12 conformation is nearly orthogonal to the active axis (Supplementary Fig. 14). Comparison of apo simulations started with helix 12 in these very different conformations shows incomplete convergence. However, both apo simulations share one low energy well/conformation indicating partial convergence (Fig. 2a, c and Supplementary Fig. 1).

These simulations indicate that the physiologic structural ensemble of apo PPARγ helix 12 is very diverse, consistent with the multiple broad peaks observed via $^{19}$F NMR (Fig. 1e). The lowest energy helix 12 conformations appear to be a mix of those favorable for binding coactivators, corepressors, or neither (Supplementary Fig. 1). In some low energy conformations, the LxxLLxxxY residues of helix 12 bind to the coregulator binding surface similar to the L/IxxIIxxxF/Y/L motif of corepressors[12]

(Supplementary Fig. 9a). Similar auto-repressed conformations have been observed in structures of other nuclear receptors, including estrogen receptor α (ERα)[46], COUP-TFII[47], testicular receptor 4[48], rat ERβ[49], retinoic acid receptor bound to an inverse agonist[50] and between PPARγ crystallographic unit cell members[24,51] (Supplementary Fig. 9b).

PPARγ LBD structures are commonly asymmetric homodimers with helix 12 in the canonical active and crystal-contact induced inactive conformation (Supplementary Fig. 14). Bonding between K347 on helix 4 and helix 12 is characteristic of this active, but not inactive, helix 12 conformation in deposited PPARγ structures (Fig. 3a). Our simulations also indicate that non active-like helix 12 conformations lack K347 to helix 12 bonding (Fig. 3b, c). Simulations started from the active, but not inactive, apo structure predict high prevalence of K347-H12 bonding (Fig. 3c). To determine the prevalence of active-like helix 12 conformations in the apo structural ensemble we introduced a K347A mutation in PPARγ and measured the effect of this

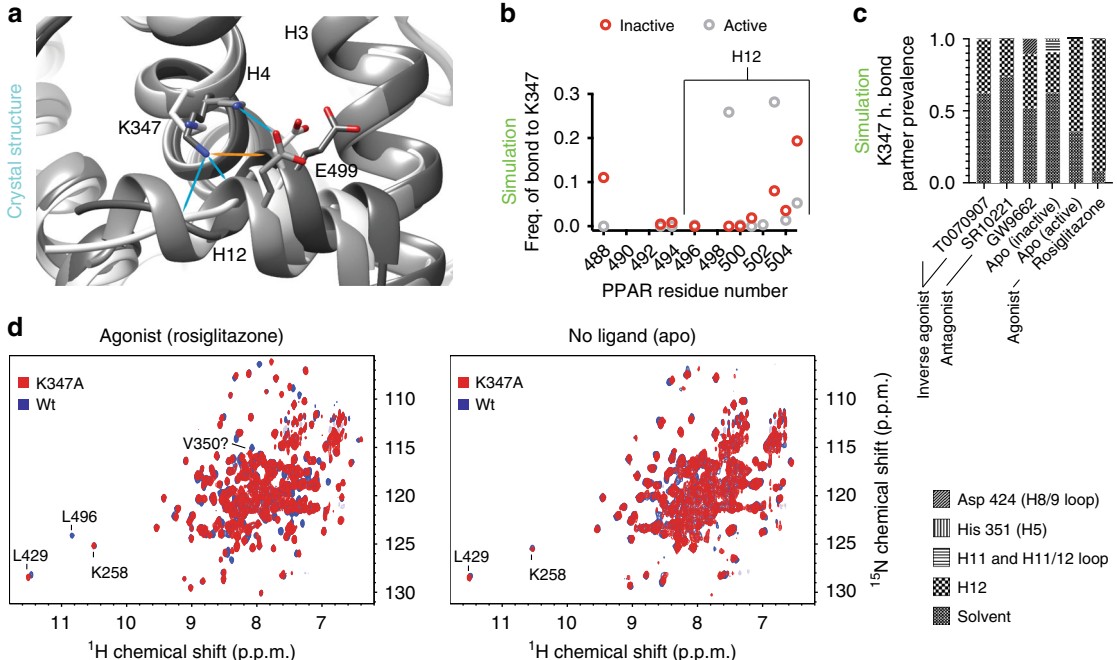

**Fig. 3 The apo ensemble contains little active-like helix 12 conformations. a** Examples of three classes of helix 4 to helix 12 bonding observed in deposited crystal structures (PDB codes 1PRG, 2PRG, and 3R5N). Almost all of the 176 active structures (chain A) available show bonding between K347 and the helix 12 backbone. Two of the 176 structures have a salt bridge between K347 and E499. None of the 176 inactive conformations (chain B) showed bonding between K347 and helix 12. **b** All 58 representative structures shown in Supplementary Fig. 1 were subjectively classified as having active-like or inactive-like helix 12 conformations. Prevalence of bonding between helix 12 residues and K347 calculated from 1 μs simulations of each representative structure in each class are shown. **c** The Boltzmann average values for interaction of the terminal side chain protons on K347 (i.e., NH3) and all other residues was calculated for PPARγ alone (apo) or bound to the indicated ligands. **d** Comparison of wt and K347A mutants bound to rosiglitazone or without ligand (apo) using TROSY-HSQC NMR. Source data are provided as a Source Data file (Source data_Heidari.xlsx).

mutation using protein NMR. If K347A impacts structure, K347 to helix 12 bonding exists, because K347 interacts with either solvent or helix 12 (Fig. 3c). As expected for an active helix 12 position, the K347A mutation induces many changes to the PPARγ-agonist (rosiglitazone) complex, including disappearance of helix 12 residue L496 and a shift in a residue near the coregulator binding surface (L429). In contrast, there are few changes to the apo PPARγ structure (Fig. 3d). This indicates that an active-like helix 12 conformation is not a major component of the apo physiologic ensemble. However, it is probable that apo helix 12 structural ensemble contains a small population of active-like conformations for the following reasons: (1) apo simulations indicate active-like states are similar energy to other states (Supplementary Fig. 1), (2) crystal structures show apo with helix 12 in an active state[40,52], and (3) there are changes in a few amide chemical shifts upon mutation of K347 (Fig. 3d).

**Structural definition of inverse agonist structural states.** To uncover physical mechanisms by which ligands induce or maintain repressive states in nuclear receptors we performed aMD of the PPARγ LBD bound to three inverse agonists/antagonists alone or bound to the corepressor NCoR and apo PPARγ LBD bound to NCoR. These simulations were started with helix 12 in the inactive conformation because of steric clash between the long CoRNR helix and helix 12 in the active conformation[53].

In order to estimate values characteristic of the solution state structural ensemble we clustered structures near the bottom of the deepest wells in the aMD generated landscape and started 1 μs conventional molecular dynamics (cMD) simulations from representative structures for each well. Cluster population weighted average values from each well were then combined

into one value using Boltzmann weighting (see methods), producing a Boltzmann average value.

Representative structures from the lowest energy wells of the PPARγ-ligand complexes (no NCoR) demonstrate considerable conformational diversity in helix 3, 11, and 12 (Fig. 2b and Supplementary Fig. 1). The two inverse agonists induce distinct low-energy helix 12 conformations (Fig. 2 and Supplementary Figs. 1 and 5) consistent with the distinct helix 12 fluorine spectrum (Fig. 1e). In addition, representative structures from the two lowest energy wells for the inverse agonist complexes feature expanded coregulator binding surfaces that could accommodate the longer corepressor helix; in contrast, helix 12 and the corepressor helix overlap in the lowest energy apo structures (Fig. 2b and Supplementary Fig. 5c). Consistent with simulation, deletion of helix 12 increased affinity of apo for corepressor peptides from SMRT and NCoR and had less or the opposite effect on the PPARγ-ligand complexes. (Supplementary Figs. 11 and 12 and Supplementary Tables 3 and 4).

We tested the prevalence of helix 12 to helix 4 bonding in these complexes, as above, using the K347A mutant. K347 mutation has large, small and almost no effect on protein NMR spectra of PPARγ bound to GW9662, T0070907, and SR10221 respectively (Supplementary Fig. 2). This indicates that the PPARγ-GW9662, PPARγ-T0070907, and PPARγ-SR10221 structural ensembles contain major, minor, and no detectable populations respectively with helix 12 in an active-like conformation. This is roughly consistent with simulations (Fig. 3c and Supplementary Fig. 1). These data suggest that an inverse agonist (T0070907) and antagonist (GW9662) share a conformation that allows helix 12 to helix 4 bonding. This conformation is likely represented by components of the right blue peak in Fig. 1e. This peak dominates the antagonist structural ensemble (88% of total), but comprises a

smaller portion of the inverse agonist ensemble (36% of total). These data also suggest that SR10221 causes a large displacement of helix 12.

For the corepressor CoRNR box peptide (NCoR) containing simulations we observed consistent bonding between PPARγ K329 (helix 3 charge clamp) and the NCoR backbone and PPARγ helix 4 residue N340 and the NCoR peptide residue R2268. We also observed salt bridge bonding between K347 (PPARγ helix 4) and primarily NCoR residue E2264 in all complexes and some, but less, salt bridge bonding between K347 and helix 12 residue E499 in all complexes except for SR10221. N340 and analogous residues bond to coactivators in PPARγ[54] and other nuclear receptors[55–57] and residues analogous to K347 were reported as important for coactivator bonding in ERα[58].

We mutated these three residues to determine effect on corepressor affinity. The effect of the K347A mutation is complicated to interpret because K347 bonds to helix 12 to varying degrees in the absence of NCoR (Fig. 3 and Supplementary Figs. 2 and 11). In contrast, mutations of K329 or N340 are likely nondisruptive[59] as they do not interact with other PPARγ residues and their NCoR bonding partners are solvent exposed in the mutants (Supplementary Table 6). Therefore, the change in NCoR affinity upon mutation of K329 or N340 to alanine should correlate exponentially with hydrogen bond prevalence to NCoR. Boltzmann average values indicate more prevalent hydrogen bonding between the NCoR peptide backbone and the helix 3 charge clamp (K329) than N340 and NCoR (Fig. 4a, b). Consistent with simulation, K329A and N340A mutations affect NCoR affinity and the K329A mutation has a larger effect than N340A (Fig. 4c).

**Ligands stabilize the helix 3 charge clamp**. We found that simulations of the helix 3 charge clamp region are consistent with the fluorine NMR in this region, which showed that all ligands stabilized this region and the most efficacious ligands did so to a greater degree. Apo PPARγ simulations, with or without NCoR peptide, show more variability in the position and helicity of the charge clamp region of helix 3 than PPARγ bound to ligands (Fig. 5a, b and Supplementary Fig. 3i, j). These multiple helix 3 conformations are consistent with the multiple wide peaks observed from a [19]F NMR of a probe placed in helix 3 charge clamp region (Q322; Fig. 1d). Exchange between helix 12 binding and unbinding to the coregulator binding surface (Supplementary Fig. 9a) could also contribute to the broad peaks in the apo helix 3 probe spectrum. This structural variability in apo helix 3 is likely functionally significant as it varies the position of the backbone of the charge clamp considerably (K329; Fig. 5c and Supplementary Fig. 3h).

**Allosteric linkage between coregulator binding and Ω loop**. The large loop connecting helix 2 and 3 is known as the omega loop (Fig. 1b). It is a structural feature common to most nuclear receptors and can be mobile because it is often unresolved in structures. Ligands can bind to a pocket partially formed by the omega loop and affect function[60]. Interactions between the omega loop and helix 3 may stabilize helix 12 in an activating position[52]. A fluorine probe on the omega loop at Q299 (Fig. 6a) produced both a narrow (72% of signal) and an overlapping wider peak (28% of signal) when PPARγ is bound to the inverse agonist T0070907 and more complex spectra for other complexes (Fig. 6). The dominant narrow peak indicates that the omega loop region is stabilized by T0070907 into a single structure in 72% of the population. Fifteen percent of the antagonist (GW9662) and apo spectra originates from a peak with very similar chemical shift and width to this narrow T0070907 peak. Addition of corepressor

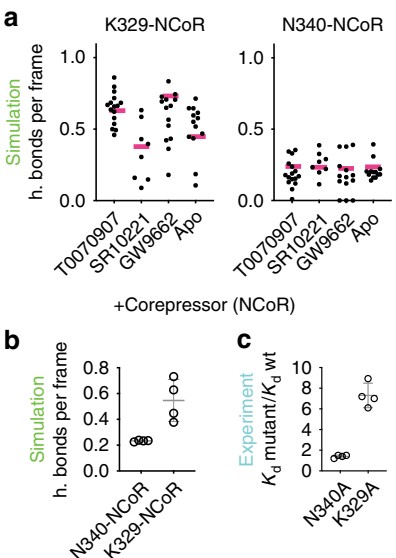

**Fig. 4 Simulation prevalence of NCoR to PPARγ binding correlates with experiment. a** The prevalence of the indicated hydrogen and salt bridge bonds in individual 1μs conventional MD runs of representative structures from low energy wells (black circles) and the overall Boltzmann weighted average (pink) for the indicated complexes. **b** Boltzmann average prevalence of hydrogen bonding between the helix 3 charge clamp residue (K329) and NCoR and between a helix 4 residue (N340) and NCoR is shown. The average and standard deviation of these 4 values are shown (0.23 ± 0.005 and 0.55 ± 0.16). $p = 9E−3$; 95% CI for difference magnitude is 0.1–0.5; $t = 3.8$; $df = 6$; unpaired two-tailed $t$ test. **c** The average decrease in affinity is greater for a mutation abolishing the helix 3 charge clamp (K329A) than for one abolishing the helix 4-NCoR bonding (N340A). The average and standard deviation of these 4 values are (1.4 ± 0.14 and 7.3 ± 1.18). $p = 6E−5$; 95% CI for difference magnitude is 4.4–7.4; $t = 9.92$; $df = 6$; unpaired two-tailed $t$ test. These data are also presented in Supplementary Fig. 11. Each open circle represents the average $K_d$ of the indicated mutant from two independent experiments divided by the average $K_d$ of wt from four independent experiments for apo and PPARγ LBD bound to T0070907, SR10221, or GW9662. Source data are provided as a Source Data file (Source data_Heidari.xlsx).

peptides drives the omega loop signal in the GW9662 and T0070907 complexes toward a single narrow peak (i.e., structure), while addition of these peptides to apo leads to less consolidation (Fig. 6 and Supplementary Fig. 4a). The other inverse agonist (SR10221) spectrum changes but does not consolidate and is very distinct from the T0070907 spectrum. The spectra of PPARγ bound to efficacious agonists (rosiglitazone or GW1929) change the least upon addition of coactivators or corepressors (Fig. 6 and Supplementary Fig. 4).

Coregulator binding induced omega loop changes demonstrate that the omega loop is allosterically linked to the coregulator binding surface in the apo, antagonist, and inverse agonist complexes. These data also indicate that a specific omega loop conformation induced by one inverse agonist (T0070907) is characteristic of a receptor-wide structural state with high affinity for both SMRT and NCoR for two reasons. First, corepressors skew the antagonist and apo omega loop structural ensemble towards narrow peaks of similar chemical shifts to this T0070907 induced peak. Second, the relative population of this narrow peak (72% of total) is similar to that of the narrow left-shifted peak in the helix 12 spectrum (64% of total).

Simulations indicate that in some low energy apo, antagonist (GW9662) and inverse agonist (T0070907) structures (with or without NCoR) helix 3 can extend n-terminally to include the

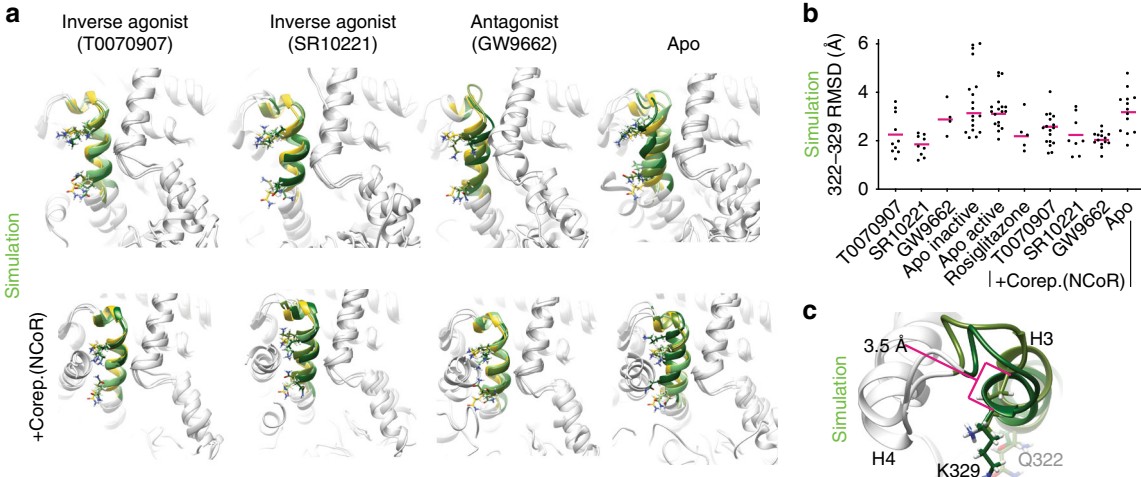

**Fig. 5 Simulations indicate that the apo helix 3 charge clamp is dynamic and is stabilized upon ligand binding. a** The c-terminal portion of helix 3 is highlighted in color. The side chains of the charge clamp (K329) and the BTFA probe location (Q322) are shown. Dark green, olive drab, and light green indicate that the structures are a representative structure from the three lowest energy wells in the aMD energy landscape for the indicated PPARγ-ligand or PPARγ/ligand/NCoR complexes. A representative structure from the only well in the rosiglitazone PPARγ simulation is shown in yellow for reference. **b** RMSD of residues 322–329 of individual cMD simulations that were started from representative structures from the lowest energy wells for the indicated complexes compared to the same residues in the crystal structure 1PRG (apo PPARγ). Boltzmann averages are shown as magenta bars. **c** Comparison of representative structures from the lowest (dark green) and second lowest (olive drab) energy wells of inactive apo PPARγ. The distance between the alpha carbon atoms for the charge clamp residue K329 for the two structures is indicated. Source data are provided as a Source Data file (Source data_Heidari.xlsx).

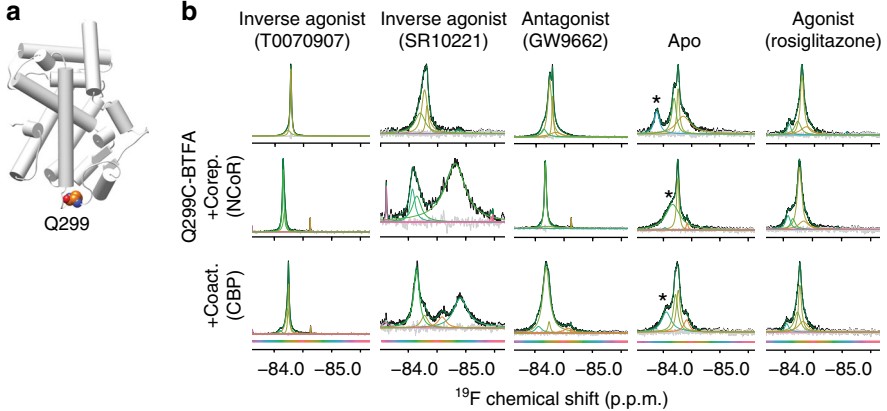

**Fig. 6 A specific omega loop conformation is induced by the inverse agonist T0070907. a** Location of the Q299 probe on the omega loop. **b** $^{19}F$ NMR spectra of PPARγ$^{Q299C}$-BTFA alone (apo) or bound to the indicated small molecules (top panel). Either corepressor (CoRNR) or coactivator (LxxLL) peptides were added to the apo and ligand complexes (lower two panels). The color of the deconvoluted peaks denotes chemical shift as shown by the color bar. Asterisk denotes signal that likely originates from the native cysteine (C313; see Supplementary Fig. 3).

omega loop probe (Q299; Supplementary Fig. 4c, d). Helix 3 extension (3–4 residues) is also found in a small minority of deposited PPARγ crystal structures (4 out of 174 structures) (Supplementary Fig. 4c, e). Helix 3 extension would be consistent with a narrow omega loop probe peak (Fig. 6b); however, Boltzmann averages of helix 3 extension (Supplementary Fig. 4c) do not correlate well with the population of the narrow peak in the omega loop spectra. In addition, HDX-MS does not show differences between apo, T0070907 and SR10221 complexes in the region of putative helix 3 extension, despite simulation differences in extension (Supplementary Fig. 8). The disagreement between simulation and NMR/HDX-MS implies that either the simulations are inaccurate or they are representative of intermediate structures.

**A salt bridge switch for inverse agonism**. We hypothesized that T0070907 conformations with helix 3 extension are representative of structural intermediates on path to an unsampled equilibrium structure with high affinity for corepressors. Simulations indicate that one consequence of helix 3 extension is reduction of the interaction between the omega loop and helix 3 (Supplementary Fig. 6a, b), which is expected to disfavor the canonical active state[38,52].

Another effect of helix 3 extension is disruption of a tripartite salt bridge (Fig. 7c and Supplementary Fig. 6h) that joins three parts of the ligand binding domain including the omega loop (E304), the adjacent helix 6–7 loop (R385) and helix 11–12 loop (E488). This network is also present in PPARα and PPARδ (Supplementary Fig. 6c). Helix 3 extension of just a few residues

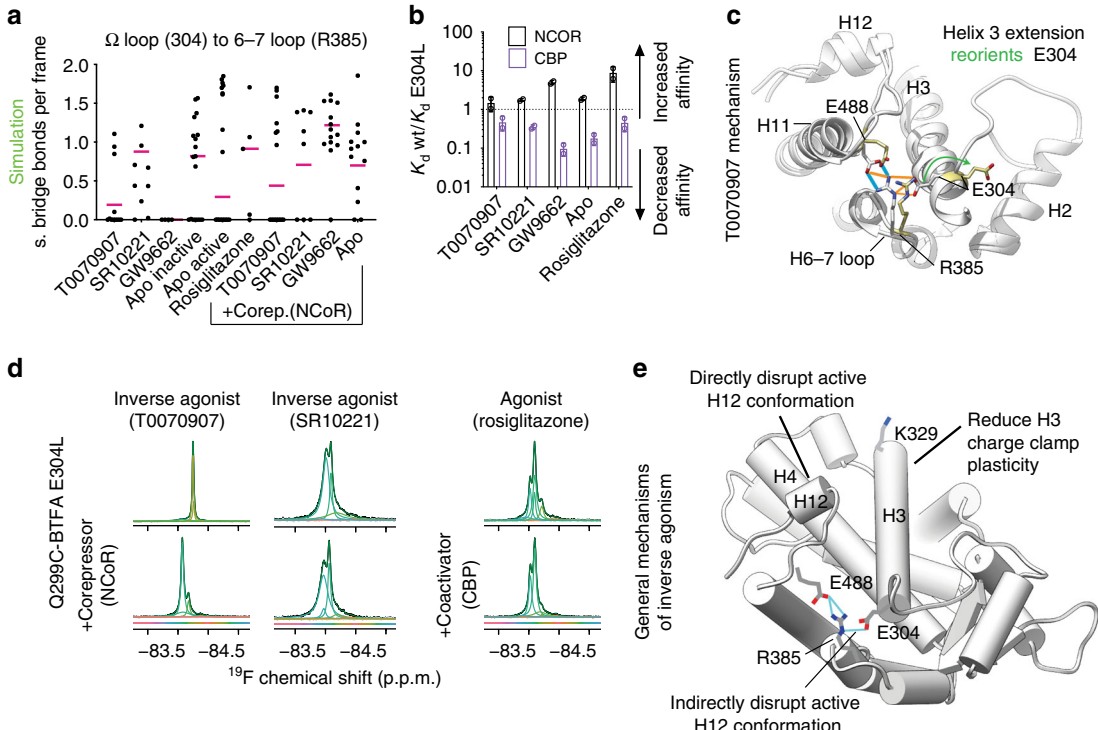

**Fig. 7 General and T0070907 specific mechanisms of inverse agonism. a** Prevalence of the helix 6–7 loop (R385) to helix 3 (E304) salt-bridge is shown for the indicated complexes for individual cMD simulations started from representative structures from the lowest energy wells in the aMD potential energy landscapes (black circles). The Boltzmann average is shown by a magenta bar. **b** The change in affinity induced by mutation E304 is shown for the indicated complexes. The average (bar heights), standard deviation (error bars) and individual values (open circles) from two independent anisotropy experiments using protein from the same purification batch are shown. **c** Proposed intermediate state induced by T0070907 as illustrated by two crystal structures (1ZEO in khaki/white and 2PRG in white) where helix 3 extension leads to reorientation of E304 and disruption of the tripartite salt bridge network that stabilizes H12 in an active conformation. This reorientation upon helix 3 extension is observed in T0070907 and other simulations (Supplementary Fig. 4c). Blue indicates ideal hydrogen bonds, while orange lines indicate relaxed constraint hydrogen bonds (20° and 0.4 Å). **d** Omega loop probe fluorine NMR spectrum of E304L (PPARγ$^{Q299C\_E304L}$-BTFA) bound to the indicated ligands with or without addition of corepressor (NCoR) or coactivator (CBP) peptide. The SR10221 used for this panel was produced at a different lab than that used in all other figures. The two batches were indistinguishable by proton NMR, mass spectrometry and their effect on affinity of PPARγ for coregulators (Supplementary Fig. 15). The color of the deconvoluted peaks denotes chemical shift as shown by the color bar. **e** The proposed general mechanisms for how inverse agonists induce higher affinity than apo for SMRT and NCoR CoRNR box peptides is shown. Source data are provided as a Source Data file (Source data_Heidari.xlsx).

beyond the usual terminus (V305) dramatically reorients E304 (E304 is usually part of the omega loop) disrupting this part of the network (Fig. 7c and Supplementary Fig. 4e).

Disruption of the tripartite salt bridge, especially the R385 to E488 bridge involving the helix 11-12 loop is expected to disrupt the active helix 12 conformation. Protein NMR shows that agonist binding reduces μs–ms motions in the tripartite salt bridge region[16,17] and mutation of R385 and E488 reduce PPARγ thermal stability[61]. Both a human PPARγ variant (F388L) associated with familial partial lipodystrophy and an R385A mutant reduce agonist potency, likely through disruption of this salt bridge network[62]. Salt bridge bonding is reduced in inactive crystal structures (Supplementary Fig. 6e).

The inverse agonist T0070907 generates the lowest Boltzmann prevalence of this salt bridge network bonding (Fig. 7a and Supplementary Fig. 6g). This raises the possibility that a key mechanism by which T0070907 induces a high affinity state for corepressors is through helix 3 extension, disrupting this network. These data led us to hypothesize that E304 is a key member of this salt bridge network that is disrupted by inverse agonists.

We tested this hypothesis using a E304L mutant. The E304L mutant increases corepressor affinity and decreases coactivator affinity across all tested complexes. However, it has a minimal effect on coregulator affinity for PPARγ-T0070907 and small effects on apo PPARγ and PPARγ bound SR10221 (Fig. 7b and Supplementary Fig. 6d). This indicates that both inverse agonists increase or maintain apo-like disruption of the E304 containing tripartite salt bridge.

We tested the consequences of E304L mutation on the equilibrium structural ensemble of the omega loop by performing $^{19}$F NMR. Comparison of omega loop signal with (Fig. 7d and Supplementary Fig. 6f) or without (Fig. 6b) the E304L mutation shows that the mutation minimally effects the T0070907 spectra, but induces larger changes in other complexes. Interestingly, the E304L mutation makes the antagonist (GW9662) and inverse agonist (T0070907) complex spectra remarkably similar (Supplementary Fig. 6f). This indicates that disruption of the tripartite salt bridge shifts the antagonist omega loop (and likely the entire protein) structural ensemble toward the inverse agonist state. The mutation also had a large impact on the spectra from complexes containing SR10221, which indicates that SR10221 solely or partially uses other mechanisms besides salt bridge disruption to accomplish helix 12 disruption and inverse agonism.

Together these functional and structural data support three ideas: (1) disruption of the tripartite salt bridge favors corepressor binding, (2) one mechanism by which an inverse agonist (T0070907) induces higher affinity than apo is through disruption of this salt bridge, and (3) the primary difference

between T0070907 and the almost chemically identical antagonist (GW9662) is efficacy of disruption of the salt bridge. In addition, these data support the idea that T0070907 and SR10221 induce repressive structural states through primarily different mechanisms.

## Discussion

Early biophysical work in myoglobin demonstrated that proteins exhibit complex dynamics on a variety of timescales[63], which has been observed in many other systems with many methods[34]. These data show that nuclear receptors are also found in several conformations of similar energies separated by kinetic barriers that result in individual state lifetimes of μs to seconds, supporting both dynamic stabilization and mousetrap-like models.

These data reveal three distinct repressive structural states for a nuclear receptor. The apo PPARγ LBD structural ensemble is distinct in helix 3, 12 and the omega loop from two inverse agonists, which in turn are different from each other. The distinct structural ensembles of helix 12, 3 and the omega loop induced by SR10221 and T0070907 lead to differential recruitment of corepressor peptides. SR10221 binding induces recruitment of a SMRT peptide but not an NCoR peptide. These structural and functional data, summarized in Supplementary Table 9, indicate SR10221 could act as a biased inverse agonist in vivo. Increased binding of SMRT, but not NCoR would be expected to produce unique functional effects in vivo as an adipocyte specific NCoR knockout emulates PPARγ agonist treatment in some respects[64]. Further structural and functional definition (in vitro and in vivo) of the effects of nuclear receptor inverse agonists is needed to clarify the structural basis and functional consequences of biased corepressor recruitment.

We propose two previously unrecognized mechanisms for induction of these distinct repressive states in PPARγ, and likely other nuclear receptors (Fig. 7e). The first mechanism is reduction of μs-ms movement in the helix 3 charge clamp region through consolidation of the ensemble to one primary structure. Many ligands (including both inverse agonists and agonists) directly interact with this region of helix 3[22,65]. Ligand induced consistent charge clamp positioning may increase coregulator affinity by minimizing the amount of binding energy invested in selecting/inducing a bonding competent conformation from the diverse apo ensemble. The second is disruption of a tripartite salt bridge that includes residues on the omega loop (E304), H6–H7 loop (R385), and the H11–H12 loop (E488). Here we found that disruption of E304 to R385 interaction via E304L mutation disrupts the active helix 12 conformation, apparently by disruption of the R385 to E488 salt bridge. Our data suggest that T0070907 disrupts the tripartite salt bridge via elongation of helix 3, which reorients E304 which also reduces helix 3-omega loop bonding. Both effects of elongation are expected to destabilize the active state[52].

In contrast to T0070907 there is no evidence of induced helix 3 elongation for the inverse agonist SR10221. SR1664, which is structurally similar to SR10221, may act as an antagonist through direct interaction with F310 near E304[21]. SR10221 could disrupt the salt bridge similarly, but more effectively. However, disruption of the salt bridge via the E304L mutation had a large effect on the SR10221 complex indicating that it may induce helix 12 destabilization via another mechanism possibly by direct interaction with helix 12 as observed in other inverse agonists[50,66] (Fig. 7e). The mechanism by which SR10221 increases affinity for one CoRNR box peptide and not another is not clear.

The presented data indicate that one inverse agonist (T0070907) induces a dominant protein wide structural state with reduced μs–ms dynamics. This major structural state of T0070907

bound PPARγ is represented by the left peak in Fig. 1e, and may remain unsampled in our simulations despite almost 30 μs of cumulative accelerated molecular dynamics simulations. Our simulations do appear to sample structures that produce the right peak in Fig. 1e. Given the slow exchange rate ($<0.4\,\mathrm{s}^{-1}$) between the left and right peaks[24], our inability to sample structures from both wells may not be surprising. This demonstrates both the limitations and the advantages of simulations for determining the mechanism of action of drugs in nuclear receptors. While some states are difficult to sample despite utilization of enhanced sampling methods, important insight can be gained from observation of ligand effects on nonequilibrium structures.

In conclusion, we detailed an undescribed mechanism for inverse agonism in nuclear receptors, that is helix 3 elongation leads to salt-bridge disruption and destabilization of the active helix 12 conformation. In addition, this work establishes the existence of compelling structural differences induced by two nuclear receptor inverse agonists that results in different corepressor peptide binding profiles. Increased understanding of the mechanisms that underlie selective nuclear receptor modulation, including ligand bias[7], helps lay the foundation for design of nuclear receptor drugs that elicit more precise functional effects in animals.

## Methods

**Protein and ligand 3D structure preparation for simulations.** All molecular complexes were prepared in silico in a similar manner to that previously described[24]. Missing residues were added using the MODELER[67] function within Chimera[68]. Complete protein structures were then run through the H++ server (http://biophysics.cs.vt.edu/H++)[69] with 50 mM salt and pH 7.4 to estimate the protonation state of ionizable amino acid side chains and were then given amber names using pdb4amber[70]. RESP charges[71] were assigned for small molecule ligands using the RED server (http://upjv.q4md-forcefieldtools.org/REDServer-Development/)[72]. Tleap[70] was used to parameterize protein and ligand with the AMBER ff14SB force field and general Amber force field (GAFF2) parameters[73–75]. The structures were immersed in an octahedron box of TIP3P[76] water molecules extended to 10 Å from the protein atoms. Enough Na+ atoms were added to neutralize the structure and KCl (K+ and Cl− ions) was added to 50 mM[77].

The crystal structure with PDB code 1PRG was used for apo active (chain A) and inactive (chain B) simulations. Rosiglitazone has two relevant protonation sites with pKas of 6.8 and 6.1 on the thiazolidinedione (TZD) and pyridine ring nitrogens[78]. The protein residues are modeled at protonation states at pH 7.4, therefore both these nitrogens were modeled as deprotonated to mimic the most likely protonation state at pH 7.4. Rosiglitazone interconverts between the R and S enantiomer with a half-life of 3 h at pH 7.2[79], and is therefore produced as a racemic mixture[78]; however, the S enantiomer has the highest affinity for PPARγ[79], and therefore the S enantiomer was modeled in our simulations based on the 2PRG crystal structure, which also has the S enantiomer. 2PRG chain A was used for the rosiglitazone build.

There is no crystal structure of SR10221[21] bound to PPARγ. To build this complex in silico we docked SR10221 into a PPARγ-SR1664 crystal structure (PDB code 4R2U) using AutoDock Vina[80]. SR10221 has no predicted pKa values near 7.4 and is an S enantiomer. The carbonyl oxygen and the carboxy group oxygens were modeled as deprotonated and the secondary amide as protonated. The overall charge was therefore modeled as −1. This model was then aligned with chain B from an apo PPARγ crystal structure (PDB code 1PRG) and the PPARγ structure from 4R2U was deleted to produce the PPARγ-SR10221 complex simulations. PPARγ from this build was aligned to apo PPARγ in the apo PPARγ-NCoR structure (see below) using Chimera. PPARγ from the apo PPARγ NCoR structure was then deleted to produce the PPARγ-SR10221-NCoR structure.

The crystal structure of GW9662-bound PPARγ (PDB code 3B0R) was used as the initial structure in all simulations of GW9662 bound PPARγ in this study. This crystal structure was also used to construct the initial 3D structure of T0070907 bound PPARγ. In this model, GW9662 was transformed to T0070907 by converting benzene ring of GW9662 to the pyridine ring. In both models, the chain B conformation was used. In order to construct the 3D structure for nuclear receptor corepressor 1 (NCoR1) bound complexes NCoR from the progesterone receptor bound to NCoR (PDB code 2OVM) was aligned to SMRT in the PPARα/SMRT crystal structure (PDB code 1KKQ). PPARα was then aligned to chain B of PPARγ alone (PDB code 1PRG) or bound to ligands (PDB codes 3B0R) and PPARα was deleted.

**Molecular dynamics simulations.** The built molecular systems were equilibrated using a nine-step of minimization and restrained simulations protocol as following. In the first step a force constant of 5 kcal mol$^{-1}$ Å$^{-2}$ was applied on the protein

heavy atoms through 2000 steps. Then, the MD simulation was performed for 15 ps with shake under constant volume periodic boundary conditions (NVT). This was followed by two rounds of 2000 steps of steepest descent minimization with 2 and 0.1 kcal mol$^{-1}$ Å$^{-2}$ spring constant. The system was then subjected to a simulation with no restraints followed by three rounds of simulations with 1, 0.5, and 0.5 kcal/mol Å$^2$ force constant on heavy atoms for 5, 10, and 10 ps. Finally, a simulation without restraints was performed for 200 ps under NPT condition. Hydrogen mass repartitioning along with SHAKE algorithm were used to allow an integration time step of 4 fs. Production MD runs of constant pressure replicates were performed from randomized initial velocities. The pressure was controlled by a Monte Carlo barostat with a pressure relaxation time (taup) of 2 ps. The Langevin dynamics with a collision frequency (gamma_ln) of 3 ps$^{-1}$ was used to keep the temperature at 310 K. The particle mesh Ewald with an 8.0 Å cutoff was used for electrostatic interaction calculations.

MD simulations were performed using two different methods: cMD and aMD. To enhance sampling some aMD simulations of molecular complexes were started from the end of previously published independent ~15-μs-long cMD production runs[24] while some were started from the equilibrated builds (see Supplementary Table 5). Average dihedral energy and total potential energy obtained from cMD simulations were used to calculate the boost parameters for aMD. In this study a dual boosting approach was carried out in which two separate boost potentials are applied to the torsional and the total potential energy terms. aMD simulations with time step of 3 fs were performed and saved every 3 ps.

All production simulations were performed using pmemd.cuda or pmemd.cuda. MPI. The simulation results were analyzed using cpptraj program in the AmberTools 14 Toolbox[81]. A toolkit of Python scripts PyReweighting was used to reweight the biased aMD frames and to calculate free energy profiles[82]. We generated the potential energy landscapes that are displayed in several of the figures in the manuscript based on aggregated accelerated molecular dynamics simulations. These energy landscapes were generated based on RMSD to residues comparisons to chain a and chain b of the PDB structure with PDB code 1PRG.

**Calculation of Boltzmann average.** In order to calculate Boltzmann averages of different structural properties of the solution state structural ensemble, the structures near the bottom of the deepest wells were searched and structures found within a 0.2 Angstrom square of the bottom of each well were clustered into 5 clusters using the k-means clustering functionality implemented in CPPTRAJ. Representative structures from the significantly populated clusters (clusters larger than 5%) were used to start 1 μs cMD simulations for each well. Using the following formula weighted average values of the given structural property for each well were calculated

$$\text{Weighted average} = w_1 x_1 + w_2 x_2 + \ldots + w_n x_n, \qquad (1)$$

where $w$ is the fraction of frames in the cluster compared to the total number of accelerated MD frames located within the 0.2 Å$^2$ centered on the bottom of the well. $x$ is the value of interest calculated from the 1 μs cMD simulation for each cluster in the well and $n$ is the number of significant clusters in each well.

Values for helicity, hydrogen bonding. etc. for each well were combined into one value for the entire energy landscape (i.e., the Boltzmann average) by multiplying the weighted average of each well by the proportion of the total protein population that would be found in each well we considered in each profile at 298 K and then summing those values across all wells we considered in the energy profile (i.e., white numbers in Supplementary Figs. 1 and 5). The relative energies of these wells are shown in Supplementary Table 7.

The proportion of protein molecules that would be expected to be found in the wells we considered (at 298 K) was calculated based using the Boltzmann relation

$$\frac{e^{\frac{E_i}{(1.9858775 \times 298 \times 0.001)}}}{\sum_0^n e^{\frac{E_i}{(1.9858775 \times 298 \times 0.001)}}} \qquad (2)$$

where $E_i$ is the energy of a given well in kcal/mol compared to the lowest energy well which is designated as 0 energy. The bottom half of the equation is the sum over all the wells considered for a given energy profile.

**Protein purification for 19F NMR.** A pET45b plasmid containing the PPARγ-LBD, residues 230–505 as well as an N-terminal 6× His tag and tobacco etch virus nuclear inclusion protease (TEV) recognition site was transformed into BL21 (DE3) gold cells (Invitrogen). Cells were grown in either terrific broth or ZYP-5052 autoinduction media. In the case of autoinduction media cells were grown for 10 h at 37 °C and then allowed to induce for an additional 12 h at 22 °C and 180 rpm. Cells in terrific broth were grown at 37 °C and 170 rpm until OD$_{600}$ of between 0.6 and 1 was reached. Following this the incubator was dropped to 20 °C for one hour and cells were induced overnight by the addition of 500 μM IPTG. Cells were pelleted by centrifugation and stored at −20 °C until ready for use. Cell pellets were resuspended in 50 mM KPO$_4$, 300 mM KCl, 1 mM TCEP, and 1 mM EDTA pH 8.0 and lysed using a C-5 Emulsiflex high pressure homogenizer (Avestin). Initial protein purification was performed on either an AKTA start (GE Healthcare) or a NGC Scout (Bio Rad) FPLC using 2 His Trap FF 5 ml columns in series (GE Healthcare). Following this the 6× his tag was removed by the addition of approximately 1:40 w/w 6× his tagged TEV and overnight incubation. Cleaved tag

and TEV protease was removed by again passing through His Trap FF columns. Size exclusion chromatography was then performed using a Hiload 16/600 Superdex 75 pg column (GE Healthcare). Protein purity in excess of 95% was confirmed by sodium dodecyl sulfate polyacrylamide gel electrophoresis.

**15N protein purification for NMR.** Similar to our previously published method of labeled protein production[24], E. coli bacteria were grown in M9 minimal media with 99% 15NH4Cl (Cambridge Isotope Laboratories) as the sole nitrogen source. For this growth, cells were grown at 37 °C in four 25 ml cultures overnight at 180 rpm. The following day, 0.5 L flasks were started with 25 mls of the overnight culture, and were incubated at 37 °C and 200 rpm until an OD600 of 1.0 then the temperature was reduced to 22 °C. After letting the cells equilibrate to the new temperature for 1 h protein expression was induced by the addition of 500 μM IPTG and cells were harvested after 16 h of additional incubation. Protein expression and purification was then done the same as for the $^{19}$F labeled protein.

**Site-directed mutagenesis.** Site-directed mutagenesis was performed using the Quikchange Lightning mutagenesis kit (Agilent Technologies). All mutations as well as the absence of spurious mutations were confirmed by Sanger sequencing (Eurofins). Primers used to generate the mutants used in this work are listed in Supplementary Table 10.

**Preparation of NMR samples.** All NMR samples were prepared to a final volume of 470 μL and a final concentration of 150 μM. Samples of PPARγ$^{C313A,Q322C}$ and PPARγ$^{Q322C}$ were loaded with ligands to a final concentration of 165 μM for all ligands except GW9662 and T0070907 which were added to a final concentration of 300 μM. Ligand concentration was varied to fix the DMSO addition to each sample at 7.8 μL, 1.66% final v/v. In the case of PPARγ$^{C313A,Q322C}$ the protein was labeled with a tenfold excess of BTFA during purification. However, the single mutant PPARγ$^{Q322C}$ was labeled with a twofold molar excess of BTFA following drug addition with the following exceptions, samples loaded with the covalent ligands were labeled with a tenfold molar excess of BTFA while the apo sample was labeled with a stoichiometric concentration of BTFA to reduce labeling on the native cysteine in the ligand binding pocket. Following labeling samples were incubated for 30 min when loaded with ligand or 2 h for the apo sample and then buffer exchanged >100× using an amicon ultra centrifugal filter (Millipore) with a 10 kDa molecular weight cut off to remove excess BTFA label. All peptides were added to NMR samples at a final concentration of 300 μM from stocks of approximately 1 mM in the same buffer as the NMR samples. Following sample preparation 10% of final volume of buffered D$_2$O was added to all samples. Samples were prepared for 2D [1H, 15N] TROSY-HSQC NMR in a similar manner, but without BTFA labeling.

**NMR spectroscopy.** Both 19F NMR and protein NMR spectra were obtained on a Bruker 700 MHz NMR with a QCI-F cryoprobe at the City University of New York Advanced Science Research Center (CUNY-ASRC) at 298.1 K. KF was set at −119.522 ppm. The trosyf3gpphsi19 and zgfhigqn.2 pulse programs were used to acquire 1D fluorine and 2D [1H,15N] TROSY-HSQC NMR data using 0.09 (1H), 0.02 (15N), and 0.83 (19F) acquisition times and typically 104 (HSQC) and 2048 (19F) scans with 2048 direct and 100 indirect dimension points (HSQC) and 65,536 points (19F). Spectral widths were 60 ppm (19F), 16 ppm (1 H), and 36 ppm (15N). Delays between scans were 1.2 s (19F) and 1 s (HSQC). Ninety-degree pulse durations were 12 μs (19F), 11–12 μs (1H) and 40 μs (15N). The variable saturating pulse location CEST experiments were carried out with a shaped pulse (Gaus1.1000, 54.52 dB, 50 ms) for a total saturating pulse duration of 1 s. The CEST experiments with variable saturating pulse duration were carried out for the indicated lengths of total saturation time with the same shaped pulse as above at on resonance and off resonance locations. The rate of exchange was fit using equation 50 from Helgstrand et al.[83] with the longitudinal relaxation rate (R$_1$) set to 2.4 s$^{-1}$ (apo PPARγ$^{Q322C}$-BTFA) and 2.7 s$^{-1}$ (PPARγ$^{Q322C}$-BTFA-GW9662-RXRα). These rates are based on measured R$_1$ for the BTFA label on apo and ligand bound PPARγ$^{K502C,C313A}$-BTFA[24]. Spectra were deconvoluted objectively with models chosen statistically by a fitting program[36]. All fits were carried out with the same settings, except where noted. Relative phase of fitted peaks was allowed to vary (π/50 rad) to accommodate imperfect phasing.

**Peptides used for NMR and HDX-MS.** The following peptides were obtained from LifeTein, LLC. Somerset, New Jersey for use in NMR and HDX-MS. All are c-terminally amidated and n-terminally acetylated (Ac) or biotinylated. CBP: biotin-Ahx-GNLVPDAASKHKQLSELLRGGSGS, MED1 Ac-VSSMAGN-TKNHPMLMNLLKDNPAQ, NCOR Ac-GHSFADPASNLGLEDIIRKALMG, SMRT Biotin-QAVQEHASTNMGLEAIIRKALMG. Ahx is a 6-carbon linker.

**Hydrogen–deuterium exchange mass spectrometry.** Hydrogen–deuterium exchange (HDX) detected by mass spectrometry (MS) Differential HDX-MS experiments were conducted as previously described with a few modifications[84]. HDX-MS samples were prepared the same as NMR samples. Peptide identification: peptides were identified using tandem MS (MS/MS) with an Orbitrap mass

spectrometer (Fusion Lumos, ThermoFisher). Product ion spectra were acquired in data-dependent mode with the top ten most abundant ions selected for the product ion analysis per scan event. The MS/MS data files were submitted to Mascot (Matrix Science) for peptide identification. Peptides included in the HDX analysis peptide set had a MASCOT score greater than 20 and the MS/MS spectra were verified by manual inspection. The MASCOT search was repeated against a decoy (reverse) sequence and ambiguous identifications were ruled out and not included in the HDX peptide set. HDX-MS analysis: For apo protein and binary complex 5 μl of sample was diluted into 20 μl D2O buffer (20 mM Tris-HCl, pH 7.4; 150 mM NaCl; 2 mM DTT) and incubated for various time points (0, 10, 60, 300, and 900 s) at 4 °C. The deuterium exchange was then slowed by mixing with 25 μl of cold (4 °C) 3 M urea and 1% trifluoroacetic acid. Quenched samples were immediately injected into the HDX platform. Upon injection, samples were passed through an immobilized pepsin column (2 mm × 2 cm) at 200 μl min$^{-1}$ and the digested peptides were captured on a 2 mm × 1 cm C8 trap column (Agilent) and desalted. Peptides were separated across a 2.1 mm × 50 mm C18 column, 3.5 μm particle size (Hypersil Gold, ThermoFisher) with a linear gradient of 4–40% CH3CN and 0.3% formic acid, over 5 min for HDX and over 1 h for data dependent acquisition (MS/ MS). Sample handling, protein digestion and peptide separation were conducted at 4 °C. Mass spectrometric data were acquired on the Fusion Lumos Oribtrap mass spectrometer (ThermoFisher). HDX analyses were performed in triplicate, with single preparations of each protein ligand complex. The intensity weighted mean m/z centroid value of each peptide envelope was calculated and subsequently converted into a percentage of deuterium incorporation. This is accomplished determining the observed averages of the undeuterated and fully deuterated spectra and using the conventional formula described elsewhere[85]. Statistical significance for the differential HDX data is determined by an unpaired t-test for each time point, a procedure that is integrated into the HDX Workbench software[86]. Corrections for back-exchange were made on the basis of an estimated 70% deuterium recovery, and accounting for the known 80% deuterium content of the deuterium exchange buffer. Data rendering: the HDX data from all overlapping peptides were consolidated to individual amino acid values using a residue averaging approach. Briefly, for each residue, the deuterium incorporation values and peptide lengths from all overlapping peptides were assembled. A weighting function was applied in which shorter peptides were weighted more heavily and longer peptides were weighted less. Each of the weighted deuterium incorporation values were then averaged to produce a single value for each amino acid. The initial two residues of each peptide, as well as prolines, were omitted from the calculations. This approach is similar to that previously described[87].

**Fluorescence anisotropy**. A 12-point 2-fold serial dilution of the respective protein loaded stoichiometrically with ligand was added to the FITC labeled peptide (50 nM). Protein concentration ranged from approximately 24–50,000 nM (exact protein range is specified in presented data). All samples were plated to a final volume of 12 μl in black low profile 384-well plates (Greiner, 784076). In the case of mutations which introduced a cysteine residue the protein was first loaded with ligand, incubated 30 min, then labeled for a further 30 min with a 2× molar excess of BTFA. Following that the samples were buffer exchanged >100× to remove excess BTFA. These samples were then plated as usual. Plates were incubated at room temperature in the dark for two hours prior to reading. All dilutions were performed in a buffer composed of 25 mM MOPS pH 7.4, 25 mM KCl, 1 mM EDTA, 0.01% fatty acid free bovine serum albumin, 5 mM TCEP, 0.01% Tween 20. Fluorescence polarization was measured with excitation at 485/20 nm and emission at 528/20 nm on a Synergy H1 plate reader (Biotek). All values reported are the average of two technical replicates. In some cases, further experimental replicates were added to allow for calculation of standard deviation. All peptides were N-terminally fluorescein labeled and had the following sequences: MED1 FITC-Ahx-NTKNHPMLMNLLKDNPAQD-NH2, NCoR FITC-Ahx-GHSFADPASNLGLEDIIRKALMG-NH2, SMRT 5FAM-HASTNMGLEAIIRKALMGKYDQW. Ahx is a 6-carbon linker (see https://www.lifetein.com/Peptide_Modifications_Pegylation_Linker.html). The MED1 and NCoR peptides were synthesized by Lifetein Inc. and the SMRT peptide was purchased from Thermo Fisher (PV4424). The Lifetein peptides are amidated on the c-terminus.

Anisotropy data were fit in Graphpad Prism 8 using an equation derived from Eqs. (6) and (40) of Roehrl et al.[88] with $A_{obs}$ (observed anisotropy) the dependent variable and $R_t$ (total receptor concentration) the independent variable

$$A_{obs} = (A_b - A_f) \frac{K_d + L_{st} + R_t - \sqrt{(K_d + L_{st} + R_t)^2 - 4L_{st}R_t}}{2L_{st}} + A_f, \quad (3)$$

where $A_b$ is the anisotropy value when the probe is bound to the receptor (i.e., at saturation), $A_f$ is the anisotropy value when the probe is free, $K_d$ is the dissociation constant, and $L_{st}$ is the total concentration of the FITC peptide (probe). Outliers were detected and eliminated from the fit automatically within Graphpad Prism using the ROUT method[89] and a Q value of 1% which specifies the maximum false discovery rate of outliers of 1%.

**Synthesis of SR10221**. A description of the synthesis and physical characterization of SR10221 synthesized at Scripps was previously published[21,90]. These methods were followed to synthesize SR10221 at the University of Montana.

See Supplementary Figs. 15 and 16 and Supplementary Table 8 for an overview of this synthesis method and quality control.

**Mass spectrometry of SR10221**. Mass spectrometry on the ligand SR10221 was performed by dissolving the ligand in 50% acetonitrile in water prior to LC-MS analysis using an Agilent 6520 QTOF coupled to an Agilent 1260 Infinity II UPLC. SR10221 was separated using a C18 reverse phase column (4.6 mm × 75 mm, 120A, Agilent) with a gradient of 50–100% ACN (0.1% formic acid) over 3 min, followed by 10 min of 100% ACN (0.1% formic acid). MS analysis was performed in high resolution mode (m/z 100–2000) with a fragmentation voltage of 80 V.

**Reporting summary**. Further information on research design is available in the Nature Research Reporting Summary linked to this article.

## Data availability
Data supporting the findings of this paper are available from the corresponding author upon reasonable request. A reporting summary for this Article is available as a Supplementary Information file.

The source data underlying Figs. 3b, c, 4a–c, 5b, 7a, b and Supplementary Figs. 3h–i, 4c, 6d, g, 7f, 8, 11, 12, and Supplementary Tables 3, 4, and 6 are provided as a Source Data file.

Raw data for the anisotropy experiments, and limited other data, are publicly available at https://osf.io/kjbam/.

HDX-MS data has been deposited in the PRIDE database with accession code PXD016401.

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

## Acknowledgements

NMR data presented herein were collected at the CUNY ASRC Biomolecular NMR Facility. We would like to thank Dr. James Aramini for his assistance in loading samples and troubleshooting technical problems. Writing assistance for this paper was provided by Shareen Grogan, Director of the University of Montana Writing and Public Speaking Center. Mass spectrometry of SR10221 was done by Eric Schultz at the UM CBSD Mass Spectrometry Core Facility. We also thank Dave Holley and the UM CBSD Molecular computation core facility for use of their computational resources. Funding for this work was provided by NIH grants R00DK103116 (T.S.H.), P20GM103546 (pilot project and J.I. grant to T.S.H.) and start-up funding provided by the University of Montana and P20GM103546 (T.S.H.). Molecular graphics and analyses performed with UCSF Chimera, developed by the Resource for Biocomputing, Visualization, and Informatics at the University of California, San Francisco, with support from NIH P41-GM103311.

## Author contributions

Z.H., I.C., and T.S.H. conceived of the experiments. Z.H. and T.S.H. performed and analyzed the simulations. I.C. and M.N. performed the NMR. I.C. performed the peptide affinity assays. P.R.G. and S.N. performed and analyzed the HDX-MS experiments. T.K., A.B., T.P., D.M., and P.D. synthesized and provided SR10221. T.S.H., Z.H., and I.C. wrote the paper with input from all authors.

## Competing interests

The authors declare no competing interests.
