## [Peer Review File · Nature Communications]

Reviewers' Comments:

Reviewer #1:

Remarks to the Author:

This manuscript addresses an important question regarding the multiple conformations of the PPAR γ ligand binding domain bound to inverse agonists and an antagonist as well as the corepressor NCoR1. Although there are over 200 crystal structures of PPAR γ bound to various ligands, no structures of PPAR γ bound to a corepressor is available. The authors combine ¹⁹F NMR of probes attached to helices 3 and 12 with accelerated MD and HDX-MS to gain insights into the conformational flexibility and multiple states of these two important helices. They also compare mutants to wild type to ascertain the importance of various postulated interactions that may favor one conformation over another. They do a great job of comparing and contrasting different data to put together a coherent model of how corepressor binding affinity is regulated by conformation of PPAR γ . Whereas the apo protein adopts a broad ensemble of structures as indicated by NMR and aMD, binding of ligands narrows the NMR peak of the ¹⁹F probe and results in a narrower distribution of states in aMD. Binding of coactivators or corepressors without an agonist or antagonist does not narrow the ensemble. Inverse agonists narrow the ensemble towards a conformation capable of corepressor binding. The antagonist, GW9662, induces the "active conformation" of helix 12 (lines 250-255). The authors tested two inverse agonists; T0070907, and SR10221 and found different perturbations in helix 12, but similar perturbations in helix 3. These results are corroborated by HDX-MS and accelerated MD analysis. The experiments appear to have been appropriately performed and analyzed and the conclusions are sound.

There are several issues with the manuscript as it is currently written.

- 1) The authors need to do a better job of making the manuscript readable for non-experts. They mix and match comparisons of different ligands rather than sticking to a consistent set. They also don't make it easy to figure out what they are studying, for example, they look at ¹⁹F probes on helices 3 and 12, but in the legend for Figure 1, it isn't until the very end that they tell you they provide a figure showing where the probes are...Why not provide those FIRST instead of LAST? The text also mixes in previous results with the current results making the results section very difficult to follow.
- 2) The authors need to do a much better job of motivating why this work is of general interest. The way the manuscript reads it seems extremely esoteric and only of interest to PPAR γ aficionados who worry about inverse agonists and the exact conformation they induce.

Reviewer #2:

Remarks to the Author:

This manuscript provides significant new insight into the mechanism by which ligands modulate the activity of the nuclear receptor PPAR γ . Early studies of PPAR γ by X-ray crystallography had led to a relatively simple explanation of activation as a basic switch mechanism. But these observations could not explain the rich behavior of this nuclear receptor in response to a range of ligands. These ligands could function as agonists, antagonists, partial agonists and inverse agonists. The simple switch could not explain this range of biological responses. Later studies involving NMR analysis were able to show that PPAR γ and its ligands populated a range of conformational states that were altered upon ligand binding.

While these NMR studies were able to demonstrate that conformational dynamics were important in the molecular mechanism, they were of limited detail. This new manuscript substantially enhances our understanding of PPAR γ . It does this particularly well by bringing a whole range of techniques to bear on the protein. These include NMR (both protein level and ¹⁹F studies of labels in several positions), HD exchange, ligand binding, and molecular dynamics. Together these methods give a much more comprehensive and believable insight than would be obtained by a single technique.

The manuscript is particularly useful in exploring the mechanism by which inverse agonists function and proposes specific structural models for this. The insights into how inverse agonists alter affinity for corepressors will be of particular interest.

Overall I strongly support the publication of this manuscript. One area that could be improved is to give additional description of the ¹⁹F 1D spectra. These are particularly important in illustrating the conformational heterogeneity. The presentation of the spectra (Fig 1, for example) shows multi-colored deconvolutions of the spectra. Given the importance of the heterogeneity observed in these spectra the authors should, at the first reference to the figures, give a brief explanation of the relationship of the deconvolution to the heterogeneity. It's apparently done with "Objective Deconvolution" (as previously published by one of the authors), but this is not likely to be familiar to most readers, and isn't mentioned in the figure legend or at first reference to the figure.

Reviewer #3:

Remarks to the Author:

This paper explores the conformational ensemble of PPAR γ . The authors delve into inverse agonism and make use of both enhanced MD simulations and NMR to explore differences between what is known from crystallography and what might be expected for a dynamic receptor. The authors have two main conclusions:

- 1) PPAR γ can adopt structurally and functionally distinct helix 3, omega loop, and helix 12 states, based on ¹⁹F NMR and enhanced MD simulations
- 2) Inverse agonism includes disruption of a tripartite salt-bridge network important to coregulator affinity

1. I think the second conclusion could benefit from ¹⁹F NMR of the double mutant in which the salt link doesn't exist (i.e. E304L, Q299C and possibly E304L,K502C).

Hopefully, with agonist and co-activator, the salt link weakening would be obvious and the role of the intermediates may also become clear with SR10221 and T0070907. I also found some of the figures confusing but I think they could be made clearer by additional annotation (ideally in the figure as opposed to the caption). Minor comments follow below:

2. The introduction of PPAR γ is a bit terse (paragraph beginning with line 46). It might help in the introduction to reference a figure pointing to helix 12,4, and 3 plus LBD, DNA binding domain, and hinge region. When complexed with a regulatory protein I gather PPAR γ is a heterodimer which then binds DNA, whereas it is a homodimer in the inactive state?

3. What do the authors mean by active state? I can imagine a series of activation intermediates associated with DNA transcription. I would also put in a description of corepressor and coactivator earlier (where the word is first used) and give the reader a sense of the focus of the paper.

4. Organization of spectra in Fig 1 is confusing. Wouldn't it make sense to group (from left to right) apo, antagonist, inverse agonist (2), and agonist and designate as such on the figure. If NCoR affinity increases with inverse agonist does CBP affinity increase with agonist? I would suggest annotating Fig 1 or the caption to remind the reader that NCoR and CBP are repressors and activators respectively.

5. Are the apo spectra monomers, dimers, or a mixture of both? is there a difference between spectra of the apo form as a function of concentration?

6. The authors write "Neither addition of coactivators nor corepressors to apo PPAR γ simplifies the

complex apo NMR spectrum. This indicates that efficacious agonists and inverse agonists induce helix 12 structural ensembles ideal for either coactivator or corepressor binding while the structurally diverse apo ensemble does not strongly favor either." Are they referring to addition of repressors and activators alone? Isn't it true that for helix 12 and helix 3 that the combination of antagonist/inverse agonist plus repressor or agonist plus activator is cooperative since the resulting spectrum is characterized more by a single resonance?

7. line 233 "corepressor helix, in contrast helix" need a semicolon

8. In the title for Figure 3 "The active apo simulation is inconsistent with NMR data" I presume they are referring to the lack of a significant active state signature as judged by the lack of change with wt to K347A? It's a confusing conclusion to me since (if I understand it correctly) the NMR data is consistent with an ensemble where the active state is weakly populated, based on Fig. 3d and the K347A mutant.

9. I'm not sure Fig 6 adds anything to the paper at least as a main figure

10. I don't understand the results of Figure 7, in particular with SR10221. is there an intuitive explanation for the dominant upfield peak with SR10221 and corepressor in combination? Is this a needed intermediate or a needed functional state?

11. E304L seems like a key mutant to perform NMR, particularly for the omega loop 19F NMR mutant . ie what does the tripartite salt link signature look like?

12. The authors write "Simulations indicate that T0070907, but not SR10221, initially disrupts the tripartite salt bridge by inducing helix 3 extension, reorienting E304 (Figure 8c)." They also write "T00709007, but not SR10221, induces an n-terminal extension of PPAR γ helix 3 which disrupts a tripartite salt bridge network near the n-terminus of helix 3 and reduces interaction of the helix 2-3 loop (i.e. the omega loop) with helix 3." Presumably this would also be reflected in 19F NMR spectra from Helix-3 and the omega loop with identical ligands? The spectra from Figure 7 with regard to the response to T0070907 and SR10221 are also very different. Presumably the E304, Q299C double mutant would distinguish the abundance of this salt link.

Author's response to reviewer comments on the manuscript titled: *Definition of functionally and structurally distinct repressive states in the nuclear receptor PPAR γ* .

We thank the reviewers for their feedback. We believe that the edits we have made and additional data we have gathered in response to this feedback will improve the readability of the manuscript and provide better support for the claims made in the manuscript. In response to several comments from several reviewers, we made many additions, edits and wording changes to the manuscript to improve readability and accessibility for non-specialists.

A request by one reviewer for additional experiments using SR10221 necessitated adding three coauthors because we had run out of our SR10221 stock obtained from Scripps and they did not have any more available. The three coauthors had synthesized and quality checked additional SR10221 for ongoing experiments, which we used for the requested experiment. SR10221 cannot be purchased and it is essential for this work.

Changes to the title, abstract and main body of the manuscript were made to fit within the journal mandated limits on length. We also added statistical data for figures and tables with significance tests.

Changes not requested by the reviewers or forced by journal conventions/word limits:

We fixed errors in **Supplementary Figure 6e** and the crystal structure data in **Supplementary Figure 6h**. We also added bonds per receptor data to panel **6e** and changed panel **6h** from fraction of receptors with bond to bonds per receptor to be more comparable to the simulation data included in the same panel.

We added three missing data points to **Supplementary Figure 6b** that were inadvertently left out. The R^2 value for **Supplementary Figure 6b** changed from 0.87 to 0.76 with the inclusion of these data points.

We changed **Supplementary figure 4e** to show the E304 side chains.

We added graphic descriptions of the small molecules used in this study to **Supplementary Figure 16**.

Supplementary Figure 5a:

The proportion of the E4 T0070907 cluster was listed as 16% it should be 14%, so this was changed.

Supplementary Table 7:

The values for apo active were wrong, we changed them to what they should be.

Supplementary Table 6:

Changed N304A-wt to N340A-wt (typo)

Figure 4: We fixed minor errors in the data in panel C.
We added **Supplementary Table 9** (a summary of results).

Reviewers' comments:

Reviewer #1 (Remarks to the Author):

This manuscript addresses an important question regarding the multiple conformations of the PPAR γ ligand binding domain bound to inverse agonists and an antagonist as well as the corepressor NCoR1. Although there are over 200 crystal structures of PPAR γ bound to various ligands, no structures of PPAR γ bound to a corepressor is available. The authors combine ^{19}F NMR of probes attached to helices 3 and 12 with accelerated MD and HDX-MS to gain insights into the conformational flexibility and multiple states of these two important helices. They also compare mutants to wild type to ascertain the importance of various postulated interactions that may favor one conformation over another. They do a great job of comparing and contrasting different data to put together a coherent model of how corepressor binding affinity is regulated by conformation of PPAR γ . Whereas the apo protein adopts a broad ensemble of structures as indicated by NMR and aMD, binding of ligands narrows the NMR peak of the ^{19}F probe and results in a narrower distribution of states in aMD. Binding of coactivators or corepressors without an agonist or antagonist does not narrow the ensemble. Inverse agonists narrow the ensemble towards a conformation capable of corepressor binding. The antagonist, GW9662, induces the “active conformation” of helix 12 (lines 250-255). The authors tested two inverse agonists; T0070907, and SR10221 and found different perturbations in helix 12, but similar perturbations in helix 3. These results are corroborated by HDX-MS and accelerated MD analysis. The experiments appear to have been appropriately performed and analyzed and the conclusions are sound.

There are several issues with the manuscript as it is currently written.

1a) The authors need to do a better job of making the manuscript readable for non-experts.

Author's response: Thank you for pointing this out, we feel the changes that we have made in relation to this request will increase the impact of this paper. We have done several things to improve readability for those unfamiliar with PPAR γ /nuclear receptor molecular pharmacology. First, we have rewritten the introduction to better frame how this structure-function work on PPAR γ fits within current nuclear receptor structure-function models and what this work adds to current models of nuclear receptor structure-function. Second, we now refer to drugs by both their name and their functional effect (inverse agonist, antagonist, agonist) both in the text and in labels on figures. Third, we refer to the probe locations throughout the manuscript in a consistent manner, which was not the case previously.

1b) They mix and match comparisons of different ligands rather than sticking to a consistent set.

*Author's response: In addition to better labeling of the ligands (see response to question 1a) all figures in the main text now show the same set of 2 inverse agonists, antagonist, apo and agonist in the same order. At times only select members of this set of 5 are shown in some figures. In the supplementary figures, there are some data from another agonist (GW1929; **Supplementary Figure 4b**) and a partial agonist (nTZDpa; **Supplementary Figure 7b**). We included these data as we felt that they added context for how the omega loop conformation is affected by coregulators when bound to inverse agonists vs. agonists and how the charge clamp region of helix 3 is affected by coregulator binding. We also included other drugs in the tables showing the effect of the labels on activity to give a fuller picture of the label's effects (**Supplementary Table 1 and 2**).*

1c) They also don't make it easy to figure out what they are studying, for example, they look at 19F probes on helices 3 and 12, but in the legend for Figure 1, it isn't until the very end that they tell you they provide a figure showing where the probes are...Why not provide those FIRST instead of LAST?

Author's response: We have rearranged Figure 1 to provide the description of probe placement first in the figure legend.

1d) The text also mixes in previous results with the current results making the results section very difficult to follow.

Author's response: We have rewritten portions of the manuscript to better separate current and previous results to make it more apparent which is which.

2) The authors need to do a much better job of motivating why this work is of general interest. The way the manuscript reads it seems extremely esoteric and only of interest to PPARg aficionados who worry about inverse agonists and the exact conformation they induce.

Author's response: We interpreted this comment as similar to that of comment 1a. See changes outlined in our response to comment 1a. In addition we have rewritten the discussion to better place this work in context of the nuclear receptor structure-function field and the protein dynamics field.

Reviewer #2 (Remarks to the Author):

This manuscript provides significant new insight into the mechanism by which ligands modulate the activity of the nuclear receptor PPAR γ . Early studies of PPAR γ by X-ray crystallography had

led to a relatively simple explanation of activation as a basic switch mechanism. But these observations could not explain the rich behavior of this nuclear receptor in response to a range of ligands. These ligands could function as agonists, antagonists, partial agonists and inverse agonists. The simple switch could not explain this range of biological responses. Later studies involving NMR analysis were able to show that PPAR γ and its ligands populated a range of conformational states that were altered upon ligand binding.

While these NMR studies were able to demonstrate that conformational dynamics were important in the molecular mechanism, they were of limited detail. This new manuscript substantially enhances our understanding of PPAR γ . It does this particularly well by bringing a whole range of techniques to bear on the protein. These include NMR (both protein level and ^{19}F studies of labels in several positions), HD exchange, ligand binding, and molecular dynamics. Together these methods give a much more comprehensive and believable insight than would be obtained by a single technique.

The manuscript is particularly useful in exploring the mechanism by which inverse agonists function and proposes specific structural models for this. The insights into how inverse agonists alter affinity for corepressors will be of particular interest.

Overall I strongly support the publication of this manuscript.

1) One area that could be improved is to give additional description of the ^{19}F 1D spectra. These are particularly important in illustrating the conformational heterogeneity. The presentation of the spectra (Fig 1, for example) shows multi-colored deconvolutions of the spectra. Given the importance of the heterogeneity observed in these spectra the authors should, at the first reference to the figures, give a brief explanation of the relationship of the deconvolution to the heterogeneity. It's apparently done with "Objective Deconvolution" (as previously published by one of the authors), but this is not likely to be familiar to most readers, and isn't mentioned in the figure legend or at first reference to the figure.

*Author's response: We have added two paragraphs to the very beginning of the results section to help the reader better interpret the ^{19}F NMR data and our deconvolution of those data. We have also added a panel to **Figure 1 (Figure 1c)** that visually describes the relationship between the ^{19}F NMR spectra appearance and the underlying energy landscape.*

Reviewer #3 (Remarks to the Author):

This paper explores the conformational ensemble of PPAR γ . The authors delve into inverse agonism and make use of both enhanced MD simulations and NMR to explore differences between what is known from crystallography and what might be expected for a dynamic receptor. The authors have two main conclusions:

- 1) PPAR γ can adopt structurally and functionally distinct helix 3, omega loop, and helix 12 states, based on 19F NMR and enhanced MD simulations
- 2) Inverse agonism includes disruption of a tripartite salt-bridge network important to coregulator affinity

1a). I think the second conclusion could benefit from 19F NMR of the double mutant in which the salt link doesn't exist (i.e. E304L, Q299C and possibly E304L,K502C). Hopefully, with agonist and co-activator, the salt link weakening would be obvious and the role of the intermediates may also become clear with SR10221 and T0070907.

*Author's response: We made the PPAR γ ^{E304L,Q299C}-BTFA mutant and report NMR spectra from this mutant bound to various drugs in **Figure 7d** and **Supplementary Figure 6f**. These two figures contain spectra from two different purification batches of PPAR γ ^{E304L,Q299C}-BTFA. The data in **Supplementary Figure 6f** appears to have more bound *E. coli* lipid, which prevented full modification with the covalent ligands, leading to some apo signal. These data added significantly to the paper. In brief, these results are consistent with the hypothesis that one of the inverse agonists disrupts the tripartite salt bridge, while the other disrupts the active helix 12 conformation through a different mechanism.*

1b) I also found some of the figures confusing but I think they could be made clearer by additional annotation (ideally in the figure as opposed to the caption).

Author's response: We have added additional annotation to most of the figures in the manuscript.

Minor comments follow below:

2. The introduction of PPAR γ is a bit terse (paragraph beginning with line 46). It might help in the introduction to reference a figure pointing to helix 12,4, and 3 plus LBD, DNA binding domain, and hinge region. When complexed with a regulatory protein I gather PPAR γ is a heterodimer which then binds DNA, whereas it is a homodimer in the inactive state?

*Author's response: We added a cartoon of the heterodimer complex on DNA showing the different domains (**Figure 1a**). In the section regarding the apo spectrum for the helix 3 NMR probe we added the following, which includes a new figure with new data (**Supplementary Figure 13**) in order to clarify whether homodimerization is likely to affect the observed spectra.*

*"Broadening is also not due to exchange between monomer and homodimer forms of apo PPAR γ . Small angle x-ray scattering and dynamic light scattering indicates PPAR γ LBD is monomeric until at least 200 μ M^{21,46}. We performed fluorescence anisotropy of labeled PPAR γ and did not detect homodimerization, furthermore helix 12 probe spectra of apo at 50 μ M, 150 μ M, and 300 μ M appears identical (**Supplementary Figure 13**)."*

We have also added the following in the introduction to clarify the roles of the heterodimer and monomer in cells.

“About half of the 48 human nuclear receptors are known to heterodimerize with RXR, including PPAR γ , however PPAR γ has been found as both monomers and heterodimers in cells leaving open the possibility that it and other RXR partners can signal as monomers¹¹.”

3. What do the authors mean by active state? I can imagine a series of activation intermediates associated with DNA transcription. I would also put in a description of corepressor and coactivator earlier (where the word is first used) and give the reader a sense of the focus of the paper.

*Author response: While it may be true that the conformation of nuclear receptors change as the transcriptional complex is formed and transcription starts, we are not claiming any knowledge of these changes. Our definition of the “active” state derives from crystal structures of PPAR γ (and other nuclear receptors) bound to coactivator peptides which show a particular conformation of helix 12. This crystal structure data is supported by previous adaptive force bias simulations. This is the best supported conformation for a nuclear receptor with high affinity for coactivators and because binding coactivators increases transcription, we refer to this conformation as the active conformation. We have added a clear illustration of this conformation and what we refer to as the “inactive” conformation in a new Supplementary Figure (**Supplementary Figure 14**).*

In the introduction this is now the first mention of the active state.

“We refer herein to the receptor conformation in coactivator LxxLL box peptide bound structures as the active state.”

4a). Organization of spectra in Fig 1 is confusing. Wouldn't it make sense to group (from left to right) apo, antagonist, inverse agonist (2), and agonist and designate as such on the figure.

*Author response: We have changed the order of results by ligand in **Figure 1** as suggested and all main Figures likewise for consistency.*

4b) If NCoR affinity increases with inverse agonist does CBP affinity increase with agonist?

*Author response: Yes, that is correct, **Supplementary Tables 1 and 2** show this. We have added descriptions to the ligand name in these tables to make this more apparent.*

4c) I would suggest annotating Fig 1 or the caption to remind the reader that NCoR and CBP are repressors and activators respectively.

Author response: We have made this change and added reminders throughout the text and other figures that NCoR is a corepressor and CBP is a coactivator.

5. Are the apo spectra monomers, dimers, or a mixture of both? is there a difference between spectra of the apo form as a function of concentration?

Author response: The apo spectra is of monomers. We have added additional references and data that support this claim. See response to comment 2.

6a). The authors write “Neither addition of coactivators nor corepressors to apo PPAR γ simplifies the complex apo NMR spectrum. This indicates that efficacious agonists and inverse agonists induce helix 12 structural ensembles ideal for either coactivator or corepressor binding while the structurally diverse apo ensemble does not strongly favor either.” Are they referring to addition of repressors and activators alone?

Author response: Yes, in these figures we add either a corepressor or a coactivator to apo PPAR γ or a drug-PPAR γ complex. We have re-written this section in an attempt to make it clearer (see excerpt below).

“We previously observed that addition of matching coregulators and drugs to PPAR γ (i.e. addition of coactivators to agonist complexes), leads to minor spectral changes for the helix 12 fluorine probe. In contrast, addition of opposing coregulators (i.e. addition of a corepressor to agonist complexes) leads to peak broadening and peak splitting. A third outcome was noted for addition of coactivators or corepressors alone to apo PPAR γ , where both had similar effects²⁸. These data indicated that efficacious agonists and inverse agonists strongly induce helix 12 structural ensembles ideal for either coactivator or corepressor binding while the structurally diverse apo ensemble does not strongly favor either.”

6b) Isn't it true that for helix 12 and helix 3 that the combination of antagonist/inverse agonist plus repressor or agonist plus activator is cooperative since the resulting spectrum is characterized more by a single resonance?

Author response: Yes, that is our interpretation also. This is what we mean by “matching” drugs and coregulators. Efficacious agonists and coactivator peptides push the receptor towards similar structural ensembles in the region being observed by fluorine NMR. Whereas efficacious agonists and repressor peptides push the receptor towards different structural ensembles, resulting in broad and/or multiple state spectra. A similar result occurs for inverse agonists and corepressors and inverse agonists and coactivators. In contrast corepressor or coactivator addition to apo has similar effects on the spectra, indication that the apo ensemble does not strongly favor binding of either.

7. line 233 “corepressor helix, in contrast helix” need a semicolon

Author response: We have corrected this.

8. In the title for Figure 3 “The active apo simulation is inconsistent with NMR data” I presume they are referring to the lack of a significant active state signature as judged by the lack of change with wt to K347A? It’s a confusing conclusion to me since (if I understand it correctly) the NMR data is consistent with an ensemble where the active state is weakly populated, based on Fig. 3d and the K347A mutant.

Author response: We have changed the title to Figure 3 to “The apo ensemble contains little active-like helix 12 conformations” for increased clarity. Hopefully it is now clearer.

9. I’m not sure Fig 6 adds anything to the paper at least as a main figure.

Author response: The HDX-MS data in what used to be Figure 6 has now been moved to Supplementary Figure 8. These data were key for determining that our simulations were representative of an intermediate state and not the final equilibrium T0070907 state.

10. I don’t understand the results of Figure 7, in particular with SR10221. is there an intuitive explanation for the dominant upfield peak with SR10221 and corepressor in combination? Is this a needed intermediate or a needed functional state?

*Author response: **Note: Figure 7 is now Figure 6.** We cannot interpret these changes well for SR10221 given our current data. We are currently working on gathering more data regarding the equilibrium state of PPAR γ bound to SR10221. This will be the subject of a future paper.*

11. E304L seems like a key mutant to perform NMR, particularly for the omega loop 19F NMR mutant . ie what does the tripartite salt link signature look like?

*Author response: We performed 19F NMR of PPAR $\gamma^{Q299C,E304L}$ -BTFA. These results are presented in **Figure 7** and discussed in the results section.*

12a). The authors write “Simulations indicate that T0070907, but not SR10221, initially disrupts the tripartite salt bridge by inducing helix 3 extension, reorienting E304 (Figure 8c).” They also write “T0070907, but not SR10221, induces an n-terminal extension of PPAR γ helix 3 which disrupts a tripartite salt bridge network near the n-terminus of helix 3 and reduces interaction of the helix 2-3 loop (i.e. the omega loop) with helix 3.” Presumably this would also be reflected in 19F NMR spectra from Helix-3 and the omega loop with identical ligands?

Author response: This would be the case if the helix 3 extension were characteristic of the primary equilibrium structural state, but not if it were an intermediate state that leads to a different primary equilibrium structural state. We distinguished between these two possibilities by comparing the 19F NMR of the omega loop and HDX-MS of the omega loop between SR10221 and T0070907. If the helical extension is a primary equilibrium structural state of the T0070907 complex then HDX-MS of T0070907 should show less exchange at the amides of the omega loop compared to the SR10221 complex because helix amides are heavily involved in hydrogen bonds, but not so much in a loop. However, HDX-MS showed no difference in amide

hydrogen deuterium exchange in this region for the two complexes. Therefore, we concluded that the helical extension was characteristic of an intermediate state induced by T0070907. The E304L peptide affinity data and the E304L-Q299C-BTFA NMR data support this conclusion.

12b) The spectra from Figure 7 with regard to the response to T0070907 and SR10221 are also very different. Presumably the E304, Q299C double mutant would distinguish the abundance of this salt link.

*Author response: **Note: Figure 7 is now Figure 6.** We made the E304L Q299C double mutant as requested and the fluorine NMR did help distinguish the abundance of the salt link. It was consistent with the hypothesis that T0070907 induces salt bridge disruption, but not SR10221. SR10221 may act primarily instead through direct interaction of its tert-butyl group with helix 12 similar to putative mechanisms proposed for other PPAR γ inverse agonists.*

*We added **Supplementary Figure 15** and **Supplementary Table 8** because we used SR10221 from a new synthesis batch for **Figure 7d**.*

Reviewers' Comments:

Reviewer #3:

Remarks to the Author:

The rebuttal and revised manuscript seems very sensible. I think the conclusions put forward in the abstract/conclusions are supported by the data and there is a nice synergy between NMR, aMD, and mutational analysis.

A few typos:

line 33 there versus their

line 44 run on sentence suggest ..."However, ..."

many places "c-terminal" versus "C-terminal"

line 165 homodimerization; furthermore

Did the authors ever examine helix-3 and 12 dynamics via normal mode analyses?

The three additional reviewer comments are italicized below.

Reviewer #3 comments:

1) The rebuttal and revised manuscript seems very sensible. I think the conclusions put forward in the abstract/conclusions are supported by the data and there is a nice synergy between NMR, aMD, and mutational analysis.

Response: We thank the reviewer for their comments.

2) A few typos:

line 33 there versus their

line 44 run on sentence suggest ..."However, ..."

many places "c-terminal" versus "C-terminal"

line 165 homodimerization; furthermore

Response: We have fixed these typos.

3) Did the authors ever examine helix-3 and 12 dynamics via normal mode analyses?

Response: We have not tried normal mode analysis. We have extrapolated coordinated movement from trajectories using principle component analysis, which has pointed to long range correlations across the entire receptor which we are including in upcoming publications. Thank you for the suggestion. It looks like tools within AmberTools can run normal mode analysis. This may provide more information regarding long-range correlations in movement in the receptor.